# Improving the Trainability of Deep Neural Networks through Layerwise Batch-Entropy Regularization

**David Peer**                                           *david.peer@deepopinion.ai*
*DeepOpinion*
*University of Innsbruck, Austria*

**Bart Keulen**                                              *bart@bartkeulen.com*
*University of Innsbruck, Austria*

**Sebastian Stabinger**                           *sebastian.stabinger@deepopinion.ai*
*DeepOpinion*
*University of Innsbruck, Austria*

**Justus Piater**                                         *justus.piater@uibk.ac.at*
*University of Innsbruck, Austria*

**Antonio Rodríguez-Sánchez**              *antonio.rodriguez-sanchez@uibk.ac.at*
*University of Innsbruck, Austria*

**Reviewed on OpenReview:** *https://openreview.net/forum?id=LJohl5DnZf*

## Abstract

Training deep neural networks is a very demanding task, especially challenging is how to adapt architectures to improve the performance of trained models. We can find that sometimes, shallow networks generalize better than deep networks, and the addition of more layers results in higher training and test errors. The deep residual learning framework addresses this *degradation* problem by adding skip connections to several neural network layers (He et al., 2016). It would at first seem counter-intuitive that such skip connections are needed to train deep networks successfully as the expressivity of a network would grow exponentially with depth. In this paper, we first analyze the flow of information through neural networks. We introduce and evaluate the *batch-entropy* which quantifies the flow of information through each layer of a neural network. We prove empirically and theoretically that a positive batch-entropy is required for gradient descent-based training approaches to optimize a given loss function successfully. Based on those insights, we introduce *batch-entropy regularization* to enable gradient descent-based training algorithms to optimize the flow of information through each hidden layer individually. With batch-entropy regularization, gradient descent optimizers can transform untrainable networks into trainable networks. We show empirically that we can therefore train a "vanilla" fully connected network and convolutional neural network—no skip connections, batch normalization, dropout, or any other architectural tweak—with 500 layers by simply adding the batch-entropy regularization term to the loss function. The effect of batch-entropy regularization is not only evaluated on vanilla neural networks, but also on residual networks, autoencoders, and also transformer models over a wide range of computer vision as well as natural language processing tasks. Our code base is available at `https://github.com/peerdavid/layerwise-batch-entropy`.

# 1   Introduction

The training of deep neural networks on large-scale datasets pushed state-of-the-art results in many different areas such as computer vision, natural language processing, time-series prediction, medicine, or drug discovery (Alassafi et al., 2022; Devlin et al., 2019; He et al., 2016; Tran et al., 2021; Xu et al., 2019). One reason for this achievement is that the expressivity of neural networks grows exponentially with the depth of the network (Raghu et al., 2017). The development and training of deep neural networks on a given task is challenging and many different methods and improvements have been developed to simplify the training of deep networks, such as a more careful weight initialization (Glorot & Bengio, 2010; He et al., 2015), better activation functions (Dahl et al., 2013), regularization methods (Ioffe & Szegedy, 2015; Srivastava et al., 2014), and innovative architectural designs (He et al., 2016; Srivastava et al., 2015; Vaswani et al., 2017). Nevertheless, designing novel architectures with high performance for different downstream tasks is a laborious one and often, cannot simply be achieved through the creation of deeper networks (Tan & Le, 2019) as at times, they may perform worse on a given task than shallower architectures or even sometimes not be trainable at all. Surprisingly for such cases, the deeper the network, the lower the accuracy (He et al., 2016), which is known as the *degradation* problem. This degradation problem can be overcome through the use of residual connections.

In this paper, we analyze the degradation problem from an information theoretical point of view. To approximate the amount of information that flows through each layer, we introduce the *batch-entropy* and show empirically that trainable networks maintain a flow of information through the network. On the other hand, whenever this flow of information collapses i.e. the batch-entropy becomes zero, the network cannot be optimized with gradient descent. Through dimensionality reduction, we found experimentally that the loss surface becomes highly non-convex which provides a hint about the reason behind the difficulty in the training of such networks (Li et al., 2018). We study this information flow collapse not only empirically, but also from a theoretical perspective and prove that classical loss functions such as the cross-entropy loss cannot be minimized with gradient descent if the batch-entropy of a single layer is zero. Based on these insights, we introduce the *batch-entropy regularization* term, which ensures a positive batch-entropy and hence enables the training of deep "vanilla" neural networks. More precisely, we extend the loss function in order to optimize not only for the task specific objective, but to also optimize the network for learnability in case it is not trainable. We will demonstrate that deep "vanilla" fully connected and convolutional networks—no skip connections, batch normalization, dropout, or any other architectural tweak or special activation function—with 500 layers can then be successfully trained. Many different normalization techniques exist that regularize the variance during forward propagation. Those techniques are mainly developed to accelerate the training of deep neural networks (Ioffe & Szegedy, 2015; Ba et al., 2016; Salimans & Kingma, 2016; Klambauer et al., 2017a). We show empirically in this paper that they can not be used to train very deep vanilla neural networks.

Another related research area studies signal propagation in random networks (Poole et al., 2016) that enabled the training of very deep networks. For example Xiao et al. (2018) developed orthogonal kernels in order to train very deep convolutional networks. While this orthogonal initialization scheme is successful in training very deep networks without skip connections, a special initialization scheme for convolutional filters with only a single non-zero entry per convolutional filter is required. In contrast, we provide a solution, inspired by information theory, where trainability is part of the objective such that untrainable networks are transformed into trainable networks during the training process. We not only show that no special initialization scheme for convolutional kernels is required, but also demonstrate the success of the proposed approach on different types of layers such as convolutional or fully connected layers.

We hope that our work will enable researchers to experiment with a much wider range of deep networks with novel layer types. To motivate this further, we study the effect of batch-entropy regularization on fully connected networks, convolutional networks, residual networks, autoencoders, as well as transformer models, for both computer vision and natural language processing tasks.

The main contributions of this paper are as follows:

- Batch-entropy is applied in order to estimate the amount of information that flows through every layer inside a network.

- A batch-entropy regularization term is introduced to enable gradient descent to optimize the flow of information through a neural network to improve trainability.

- We demonstrate that deep, vanilla, fully connected, as well as convolutional networks can be trained with batch-entropy regularization.

- We evaluate the effect of the proposed approach on fully connected networks, convolutional networks, residual networks, autoencoders, as well as transformer models, for both computer vision and natural language processing tasks.

Related work is given in section 2. In section 3 we introduce the batch-entropy to approximate the amount of information that flows through a network and the batch-entropy regularization which enables gradient descent to optimize the flow of information through each layer. In section 4, we experimentally analyze the positive effects of batch-entropy regularization. We finish our paper with a discussion on our findings and propose future work in section 5.

## 2 Related Work

Optimization problems in deep neural networks have been the subject of a wide range of studies. These studies have shown that a careful initialization of parameters has a significant effect on the training (LeCun et al., 2012) and revealed the problem of vanishing or exploding errors and how to keep the error flow constant (Hochreiter & Schmidhuber, 1997). Initialization methods—that initialize weights such that gradients are not vanishing or exploding at the beginning of the training—have been the subject of extensive study (Glorot & Bengio, 2010; He et al., 2015; Hinton et al., 2006; Krähenbühl et al., 2016; Mishkin & Matas, 2016). Unfortunately, deep neural networks are still difficult to optimize, even when those initialization methodologies are used (Srivastava et al., 2015). He et al. (2016) called this the degradation problem and showed that special architectures with skip connections (Srivastava et al., 2015; He et al., 2016) could avoid this problem. Later, Peer et al. (2021b) found that certain layers in neural networks exist that may be the source of the degradation in performance of very deep models and proved that skip-connections would bypass those misleading layers. The same authors showed that those layers do not contribute to the classification and therefore, around 60% of the layers of a trained residual network can be pruned with a negligible effect on the test-error (Peer et al., 2021a).

Many different normalization techniques exist that regularize the variance during forward propagation. Batch normalization accelerates the training by reducing shifts of output distributions of hidden layers (Ioffe & Szegedy, 2015). The effect of batch normalization strongly depends on the batch-size. Therefore, Ba et al. (2016) introduced layer normalization which uses the mean and variance from all of the summed inputs in order to reduce the training time. Weight normalization decouples the length of weight vectors from their direction in order to accelerate the training of deep networks (Salimans & Kingma, 2016). Deeper networks can indeed be trained with self normalizing networks Klambauer et al. (2017a), but they do not work on vanilla feedforward neural networks with ReLU activation functions. They implement a special scaled exponential linear unit in order to achieve training of networks with 32 layers as was demonstrated empirically in the original paper.

Although these techniques accelerate the training of deep networks, they do not improve the trainability of very deep vanilla networks as will be shown later in this paper. Another research area studies this problem by analyzing signal propagation in neural networks through mean field theory. The theoretical framework by Poole et al. (2016) revealed a maximum depth which signals can propagate through at initialization for fully connected networks. This was extended by Xiao et al. (2018) to derive a mean field theory for signal propagation in random convolutional neural networks. The authors specifically introduced orthogonal kernels for convolutional layers in order to train very deep vanilla convolutional networks. Martens et al. (2021) introduced deep kernel shading to train deep neural networks without skip connections or normalization layers, but the authors use a combination of precise parameter initialization, activation function transformations

as well as small architectural tweaks together with the second-order K-FAC optimizer (Martens & Grosse, 2015) to achieve this.

Other authors have studied neural networks from an information theoretical viewpoint to improve their performance: Blot et al. (2018) proposed a regularizer in order to improve the intermediate representations of single samples $x$ by calculating the entropy of a sample conditioned to the corresponding label $y$. The authors showed that a model that is invariant to many transformations will produce the same representation for different inputs, which improves the performance of the model. Achille & Soatto (2018) showed that a representation can be made invariant against nuisances—random variables that affect the observed data, but are not informative for the task—by limiting its information throughput. This can be achieved by injecting noise (e.g. dropout) or creating a bottleneck (e.g. a pooling layer) through a regularization term which penalizes the information that is contained in the weights. Gilad-Bachrach et al. (2003) studied the tradeoff between the complexity of data representation and the accuracy of the model. Another information theory based regularization method is proposed by Pereyra et al. (2017), which penalizes low entropy output distributions which acts as a strong regularizer in supervised learning, improving the overall performance of the model. Upper bounds on the generalization error of learning algorithms based on an information-theoretic analysis have also been derived (Xu & Raginsky, 2017).

## 3 Methods

In this section, the flow of information through neural networks is analyzed from a theoretical perspective. We first introduce the batch-entropy to measure the amount of information that is propagated through each individual layer. We then prove that a positive batch-entropy is required at each layer in order to successfully optimize a given objective (e.g. the cross-entropy between the output and the ground truth) with gradient descent. Based on those insights we introduce a regularization term that improves trainability.

### 3.1 Notation

Scalar values are represented with lower case letters such as $k$, vectors are written in bold $\mathbf{x} = [x_1, x_2, ...x_k]^T$ and matrices are represented with bold upper case letters $\mathbf{W}$. Sets of vectors are represented by $\mathcal{X} = \{\mathbf{x}_1, \mathbf{x}_2, ...\mathbf{x}_k\}$.

We call a training set $\mathcal{S} \in \mathcal{X} \times \mathcal{Y}$ which contains items (e.g. images) $\mathcal{X}$ and their labels $\mathcal{Y}$ [1]. We assume that the ground truth $\mathbf{y} \in \mathcal{Y}$ is one-hot encoded. $c$ is the dimensionality of a single label. The input to a layer is of dimension $m$ and its output dimension $n$. Weight matrices $\mathbf{W} \in \mathbb{R}^{n \times m}$ and bias terms $\mathbf{b} \in \mathbb{R}^{n \times 1}$ are trainable parameters of a network. The outputs of a given layer $l \in \{1, ..., L\}$ for a neural network with $L$ layers are vectors $a^{l+1}(\mathbf{x}) = f(\mathbf{z}^{l+1})$ with $\mathbf{z}^{l+1} = \mathbf{W}^l a^l(\mathbf{x}) + \mathbf{b}^l$ computed for a given input $\mathbf{x} \in \mathcal{X}$ with a nonlinearity function $f$ and $a^0(\mathbf{x}) = \mathbf{x}$. The output of an individual neuron $i$ at the output for layer $l$ would be $a_i^l(\mathbf{x})$.

The network is trained with mini-batches that satisfy $\mathcal{B} \subseteq \mathcal{X}$. Without loss of generality it is assumed that batches $\mathcal{B}$ are uniformly distributed w.r.t. the class labels. The surrogate loss, optimized with gradient descent, is calculated using the cross-entropy loss as $\mathcal{L}(\mathbf{x}) = \mathcal{L}_{CE}(g(\mathbf{x}), \mathbf{y})$ for input $\mathbf{x}$ with corresponding label $\mathbf{y}$, where $g(\mathbf{x}) = \text{softmax}(\theta)$ with $\theta = \mathbf{W}^L a^L(\mathbf{x}) + \mathbf{b}^L$. The gradient is then

$$\frac{\partial \mathcal{L}}{\partial \mathbf{W}^L} = \frac{\partial \mathcal{L}}{\partial g} \frac{\partial g}{\partial \theta} \frac{\partial \theta}{\partial \mathbf{W}^L} = (g(\mathbf{x}) - \mathbf{y}) \, a^L(\mathbf{x})^T.$$

The gradient for the whole mini-batch can be calculated as

$$\frac{\partial \mathcal{L}(\mathcal{B})}{\partial \mathbf{W}^L} = \frac{1}{|\mathcal{B}|} \sum_{b=1}^{|\mathcal{B}|} \frac{\partial \mathcal{L}(\mathbf{x}_b)}{\partial \mathbf{W}^L}.$$

---

[1]For simplicity we put our focus on classification problems although the theory can be extended to e.g. regression problems.

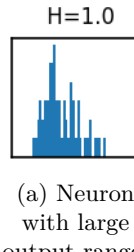
H=1.0

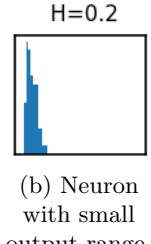
H=0.2

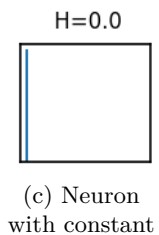
H=0.0

(a) Neuron
with large
output range.

(b) Neuron
with small
output range.

(c) Neuron
with constant
output values.

Figure 1: Histogram of 1024 output values of three different neurons from a fully connected network trained on MNIST. The x-axis shows the output-value and the y-axis how often the neuron fired with this value. On the top of each plot, the estimated batch-entropy eq. (3) is shown. More neurons are shown in appendix A.

### 3.2 Quantifying the Flow of Information through each Layer

In order to analyze the flow of information through neural networks during forward propagation, we quantify the amount of information that is propagated through each individual layer as the average amount of information that is propagated through *each individual neuron* within a layer. Output values of neurons are continuous and therefore, the amount of information passed through a neuron can be calculated using the *differential entropy* (Shannon, 1948):

$$h_i^{(l)} = -\int_{\infty}^{\infty} a_i^l(x) \ \log a_i^l(x) \ \ dx. \tag{1}$$

Unfortunately, the underlying data-generating distribution of individual neurons is unknown. Nevertheless, we have access to samples of the distribution by measuring output values of neurons for different inputs of our training dataset $\mathcal{X}$. Based on those samples, the entropy of each neuron can be approximated with e.g. nearest neighbor distances (Beirlant et al., 1997). We make use of the batch-entropy to regularize neural networks with gradient descent, which requires a differentiable batch-entropy. Nearest neighbor methods provide good approximations of the real entropy but are not differentiable. Hence, a batch-entropy using nearest neighbor methods is not suitable as a regularizer. Therefore, we search for an *approximation of the entropy that is differentiable and can be used in combination with a gradient descent based optimizer.*

To develop a differentiable function that approximates the entropy, we analyzed the output values of different, randomly selected neurons from a fully connected network. Histograms for three randomly selected neurons sampled from a neural network trained on MNIST are shown in fig. 1. More neurons for detailed analysis are shown in appendix A. It can be seen that output values of almost all neurons seem to be Gaussian distributed. Using the empirically motivated assumption that neurons are Gaussian distributed, the following differentiable approximation of the entropy can be derived:

$$H = \frac{1}{2}\log(2\pi e \ \sigma_k^2 + \epsilon), \tag{2}$$

where $\sigma_k$ is the standard deviation of a single neuron $k$, computed for the values of a mini-batch $\mathcal{B}$. A mathematical derivation of the differential entropy for the normal distribution is given by Michalowicz et al. (2013). A bias term of $\epsilon = 1$ was introduced in all our experiments to not only ensure numerical stability, with $\epsilon = 1$ we also ensure that $H \in [0, \infty)$.

Figure 1 shows how entropy values ($H$) correlate with the output distribution: For example, the neuron shown in fig. 1b has a smaller batch-entropy than the neuron shown in fig. 1a because the latter neuron fires values in a much larger range and hence propagate more information to the next layer. In the extreme case shown in fig. 1c, where a neuron outputs the same constant value for each input, the entropy is zero, which is reasonable as each input leads to the same output.

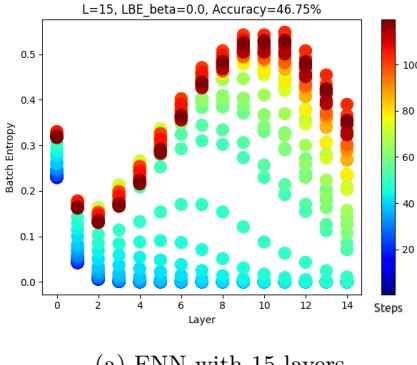
(a) FNN with 15 layers.

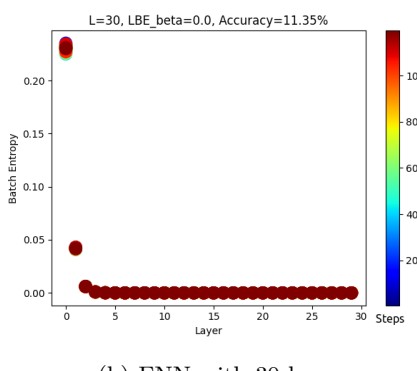
(b) FNN with 30 layers.

Figure 2: Flow of information through different layers of two fully connected networks (FNN) with 15 layers (a) and 30 layers (b).The $x$ axis shows the layer $l$ of the network which is evaluated and the $y$ axis is the corresponding batch-entropy. The color scheme indicates the step after which the batch-entropy was evaluated. Early training stages are blue, later training stages are shown in red. At the beginning of training (blue), the batch-entropy decreases as we go up in the network hierarchy (from left to right).

The entropy of a complete layer $l$ with $n$ neurons can be calculated by averaging the entropy of all neurons as follows:

$$H^l = \frac{1}{2n} \sum_k^n \log(2\pi e \ \sigma_k^2 + \epsilon), \tag{3}$$

where $\sigma_k$ is the standard deviation of neuron $k$ of layer $l$ computed for the values of the mini-batch $\mathcal{B}$. We call eq. (3) the *batch-entropy* of layer $l$.

By computing the batch-entropy (eq. (3)), we can measure the flow of information through a network, so that two different networks can easily be compared in terms of how information propagates through them. An example of such a comparison is shown in fig. 2 for a network with 15 layers that is trainable, and a network with 30 layers that is untrainable using gradient descent. Both networks are initialized with the method proposed by He et al. (2015) and trained on MNIST for 2 epochs. Figure 2a shows that the shallow network was successfully trained on MNIST. At the beginning of the training (blue dots), the flow of information is high in the early layers and decreases as we go deeper in the network. After analyzing the batch-entropy at the output layer, we found that it is slightly larger than zero, which would indicate that at least some useful information is propagated from the first to the last layer. During training, the information flow increases also for later layers (fig. 2a, color change from blue to red). On the other hand, fig. 2b shows the case of a deeper network where no information is propagated to the output layer. After testing many different hyperparameter settings, we found that such networks cannot be trained at all and the information flow never increases during training. Next, we analyze from a theoretical viewpoint why neural networks are not trainable whenever $\exists \ l \in \{1, ..., L\}$ such that $H^l = 0$ :

First of all, whenever $H^l = 0$, then for all subsequent layers $k > l$ every input of a mini-batch is the same such that $\forall k > l, H^k = 0$ which implies that $H^L = 0$. An information theoretic proof of this statement is given in appendix B. Therefore, it is sufficient to prove that a network is untrainable whenever $H^L = 0$. We need to show next that the gradient points into a direction that is independent of the desired labels $\mathbf{y} \in \mathcal{Y}$ whenever $H^L = 0$. A gradient that is independent of its ground truth cannot update weights to improve the performance which will therefore conclude our statement.

Without loss of generality we can assume that labels are one-hot encoded vectors $\mathbf{y}$ that are uniformly distributed w.r.t different classes in our mini-batch $\mathcal{B}$. It follows that

$$\frac{\partial \mathcal{L}(\mathcal{B})}{\partial \mathbf{W}^L} = \frac{1}{|\mathcal{B}|} \sum_{b=1}^{|\mathcal{B}|} \frac{\partial \mathcal{L}(\mathbf{x}_b)}{\partial \mathbf{W}^L} = \cdots = \left( \mathbf{g} - \frac{1}{c} \ \mathbf{1} \right) \ \mathbf{a}^T \tag{4}$$

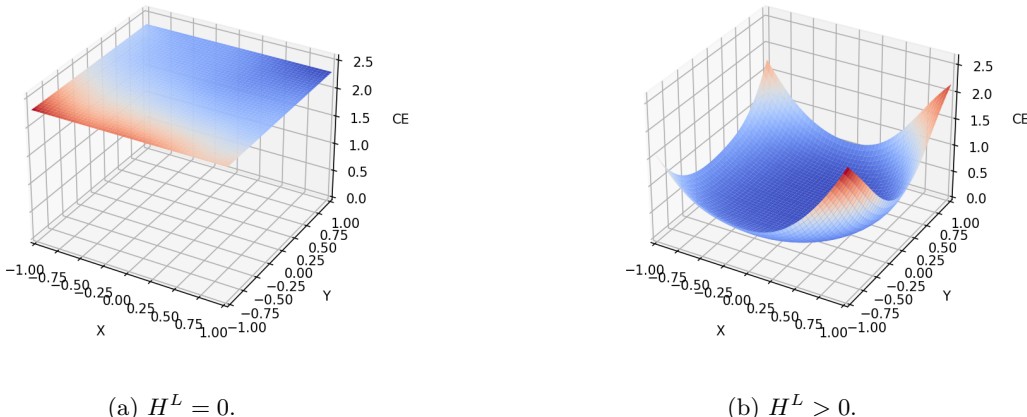

(a) $H^L = 0$.  (b) $H^L > 0$.

Figure 3: Comparison of the cross-entropy loss ($\mathcal{L}_{CE}$) surface for networks with different information through-put trained on MNIST. $X$ and $Y$ represent two random orthogonal directions in the weight space (Li et al., 2018).

where $\mathbf{1}$ is a vector with components 1 of dimension $c$ for a dataset consisting of $c$ classes. A detailed derivation of the gradient is added to appendix C.

It can be seen that the gradient is independent of labels $\mathbf{y}$ whenever $H^L = 0$. Therefore, the error signal is invalid w.r.t the real objective, and weights are updated in misleading directions. Whenever the output $\mathbf{g}_i$ of neuron $i$ is $\mathbf{g}_i > \frac{1}{c}$ weights are updated in order to decrease the output in the next iteration. On the other hand, whenever $\mathbf{g}_i < \frac{1}{c}$ weights are updated in order to increase the output value. Therefore, during the training, weights are adjusted until each output neuron fires with a constant value of $\frac{1}{c}$. Whenever $\mathbf{g} = \frac{1}{c} \mathbf{1}$ the gradient becomes $\frac{\partial \mathcal{L}(\mathcal{B})}{\partial \mathbf{W}^L} = 0$. It can then be concluded that the network fires with constant values and the performance of the model accuracy can no longer increase, as the gradient is zero.

We analyzed the output values of neurons for networks with $H^L = 0$ and have indeed found that those fire with a constant value of $\frac{1}{c}$ after a few training steps. Additionally, we evaluated whether the gradient becomes zero whenever $H^L = 0$ by evaluating the loss surface for both cases, $H^L > 0$ and $H^L = 0$. The loss surface of the cross-entropy in the weight space is shown in fig. 3. We generated two random orthogonal vectors to plot a 3-dimensional loss surface using the methodology proposed by Li et al. (2018), more samples are shown in appendix E. Networks are easier to optimize whenever the loss surface seems close to being convex in this highly compressed space (Li et al., 2018). We can confirm this finding as the surface for $H^L = 0$ (fig. 3a) is flat, highly non-convex and that the network was not trainable at all. Additionally, the gradient is zero in this region which would explain why gradient descent cannot find a solution. On the other hand, whenever $H^L > 0$ (fig. 3b) we found that the network can be trained successfully and that the test accuracy increased during training. The loss surface of the highly compressed space as shown in fig. 3b looks convex and according to Li et al. (2018), the network is therefore easier to optimize. This theoretical and empirical study leads to the following hypothesis:

**Hypothesis 1.** A network is not trainable with gradient descent, if $\exists \, l \in \{1, ..., L\}$ such that $H^l = 0$ for a network of depth $L$.

Next we provide a solution to escape from regions where $H^L = 0$ in the form of a regularization term that will help gradient descent move into regions where $H^L > 0$. This solution will allow at-first untrainable networks to be trained by bending the loss surface such that the gradient points into a direction where the cross-entropy loss can be optimized successfully.

### 3.3 Batch-Entropy Regularization

We introduce a regularization term $\mathcal{L}_{BE}$ that allows gradient descent to directly optimize the batch-entropy within each layer. Thanks to this analysis of batch-entropy, weights of layers can be adjusted such that information flows through the whole network converting invalid error signals of the cross-entropy into valid error signals.

The batch-entropy is differentiable and can be used directly as part of the loss term. Unfortunately, we found empirically that minimizing the cross-entropy and maximizing the batch-entropy leads to networks with low performance. This is reasonable because $H^l \in [0, \infty)$. Because of this, the entropy can always be maximized while the cross-entropy loss could be neglected, leading to sub-optimal networks with low accuracy. This problem can be solved by defining the precise level of information that should be passed through each individual layer. We do so through a parameter $\alpha^l$ per layer, which defines the desired level of information that should pass through each layer. Therefore, during the training, weights of each layer $l$ can be adjusted until $H^l = \alpha^l$.

We showed in fig. 2 that the information flow is different for each layer. Unfortunately, we were not able to postulate a theoretical framework that would describe for each layer $l$ which exact $\alpha^l$ value would lead to high generalization capabilities for a specific model. Figure 2a provides us with a hint that the information flow has different compression and expansion phases. Instead of having to specify a precise value for $\alpha^l$, we include it as part of the training process. Note that we pre-initialize each $\alpha^l$ with the value $\alpha$ at the beginning of the training. We found $\alpha$ through a hyperparameter grid search for each architecture individually. Additionally, $\alpha^l$ is constrained to $\alpha^l \in (\alpha_{min}, \infty)$ with $\alpha_{min} > 0$, ensuring a flow of information through the network. For $\alpha_{min}$ we found that it is sufficient to set it to a small positive value (e.g. 0.2). The *layerwise batch-entropy loss* for a single layer $l$ can therefore be calculated with

$$\mathcal{L}_{BE}^l = \left( H^l - \max(\alpha_{min}, |\alpha^l|) \right)^2 . \tag{5}$$

The batch-entropy loss ($\mathcal{L}_{BE}$) for the whole network can be calculated as follows:

$$\mathcal{L}_{BE} = \frac{\beta}{L} \sum_{l=0}^{L} \mathcal{L}_{BE}^l \tag{6}$$

The number of additional learnable parameters is $L$ (one $\alpha^l$ for each layer) which is negligible for networks with millions of parameters. In order not to exceed the cross-entropy loss ($\mathcal{L}_{CE}$) and be able to focus on the $\mathcal{L}_{BE}$ whenever the cross-entropy cannot be optimized, we included a $\beta$ hyperparameter which we find through a hyperparameter search for each experiment individually. Additionally, we scale the $\mathcal{L}_{BE}$ with the $\mathcal{L}_{CE}$ which ensures that optimizing the information flow is secondary and mainly active whenever the $\mathcal{L}_{CE}$ is large i.e. in a local minimum:

$$\mathcal{L} = \mathcal{L}_{CE} + \mathcal{L}_{CE} * \mathcal{L}_{BE}. \tag{7}$$

The effect of scaling the $\mathcal{L}_{BE}$ with $\mathcal{L}_{CE}$ can be analyzed by computing the gradient:

$$\frac{\partial \mathcal{L}}{\partial \mathbf{W}^l} = \frac{\partial \mathcal{L}_{CE}}{\partial \mathbf{W}^l} + \frac{\partial (\mathcal{L}_{CE} * \mathcal{L}_{BE})}{\partial \mathbf{W}^l} = \frac{\partial \mathcal{L}_{CE}}{\partial \mathbf{W}^l} + \mathcal{L}_{CE} \frac{\partial \mathcal{L}_{BE}}{\partial \mathbf{W}^l} + \mathcal{L}_{BE} \frac{\partial \mathcal{L}_{CE}}{\partial \mathbf{W}^l}. \tag{8}$$

Whenever the cross-entropy is small, the gradient of the $\mathcal{L}_{BE}$ becomes negligible and the main objective—$\mathcal{L}_{CE}$—is optimized. On the other hand in a local minimum where the $\mathcal{L}_{CE}$ is large, the gradient of the $\mathcal{L}_{BE}$ is additionally increased. We now continue our analysis for the case where the cross-entropy is in a local minimum:

$$\frac{\partial \mathcal{L}}{\partial \mathbf{W}^l} = 0 + \mathcal{L}_{CE} \frac{\partial \mathcal{L}_{BE}}{\partial \mathbf{W}^l} + 0 = \cdots = \mathcal{L}_{CE} \frac{\beta}{L} \frac{\partial \mathcal{L}_{BE}^l}{\partial \mathbf{W}^l}. \tag{9}$$

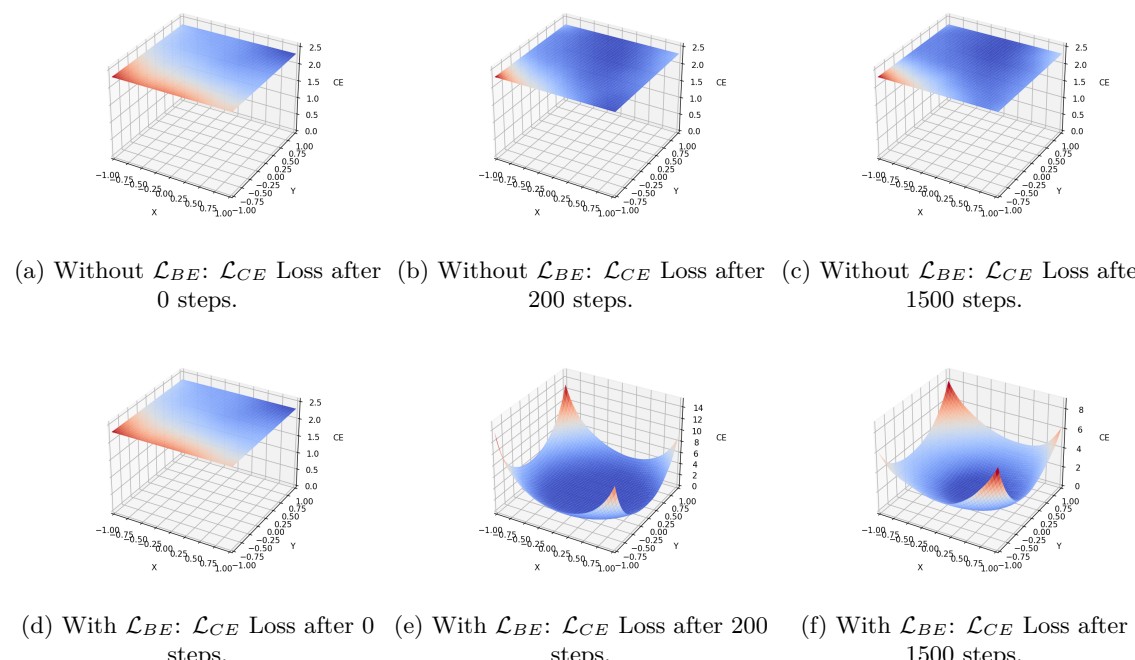

(a) Without $\mathcal{L}_{BE}$: $\mathcal{L}_{CE}$ Loss after 0 steps.   (b) Without $\mathcal{L}_{BE}$: $\mathcal{L}_{CE}$ Loss after 200 steps.   (c) Without $\mathcal{L}_{BE}$: $\mathcal{L}_{CE}$ Loss after 1500 steps.

(d) With $\mathcal{L}_{BE}$: $\mathcal{L}_{CE}$ Loss after 0 steps.   (e) With $\mathcal{L}_{BE}$: $\mathcal{L}_{CE}$ Loss after 200 steps.   (f) With $\mathcal{L}_{BE}$: $\mathcal{L}_{CE}$ Loss after 1500 steps.

Figure 4: Loss surface of a fully connected network at different training steps (0, 200 and 1500) when $H^L = 0$ at the beginning of the training without $\mathcal{L}_{BE}$ (a, b, c) and including $\mathcal{L}_{BE}$ regularization (d, e, f). Compare the errors surfaces in the top row (without $\mathcal{L}_{BE}$) with the ones in the bottom row (with $\mathcal{L}_{BE}$)

A detailed derivation is given in appendix C.

Whenever the cross-entropy cannot be optimized because $H^l = 0$, gradient descent can still optimize weights of each layer to ensure that $H^l > 0$. We additionally verify this statement empirically in section 4. A graphical illustration of why optimization without the $\mathcal{L}_{BE}$ fails and why it works with batch-entropy regularization is shown in appendix D. Without batch-entropy regularization, the $\nabla\mathcal{L}_{CE}^l = 0$ is propagated backwards through the whole network. With batch-entropy regularization, the $\mathcal{L}_{BE}$ gradient is directly applied to the weights of each layer ensuring that $H^l > 0$ as the training proceeds.

We empirically evaluated the loss surface during training as shown in fig. 4 and compared the $\mathcal{L}_{CE}$ loss after different training steps when $\mathcal{L}_{BE}$ is used, to further support hypothesis 1. It can be seen that the $\mathcal{L}_{BE}$ loss pushes weights into a direction where the cross-entropy loss can be successfully optimized with gradient descent, while networks that are trained without $\mathcal{L}_{BE}$ are unable to escape from local minima. We report more loss surfaces after different stages of training in appendix E, where we provide the cross-entropy loss surface, the $\mathcal{L}_{BE}$ loss surface, and the accuracy surface.

In fig. 2b we have shown that networks with $H^L = 0$ are not trainable. We executed this experiment again, but added the batch-entropy regularization term to the loss with $\alpha = 0.5$ and $\beta = 0.2$. The results are shown in fig. 5. It can indeed be seen that the network with 30 layers that was not trainable at first (fig. 2b) became trainable (fig. 5b), so that the accuracy increased from $11.35\%$ (fig. 2b) to $42.95\%$ in only 100 steps of training. In this same figure, we can see that the network that was trainable before (fig. 2a) is not negatively influenced by the batch-entropy regularization term. The accuracy is even a bit larger indicating that the $\mathcal{L}_{BE}$ can have a positive effect on the training of networks that are trainable without $\mathcal{L}_{BE}$ too. Therefore, we hypothesise that

**Hypothesis 2.** Regularizing the information flow through neural networks can improve the performance of the trained model.

We study hypothesis 1 and hypothesis 2 empirically in section 4 on different architectures and tasks.

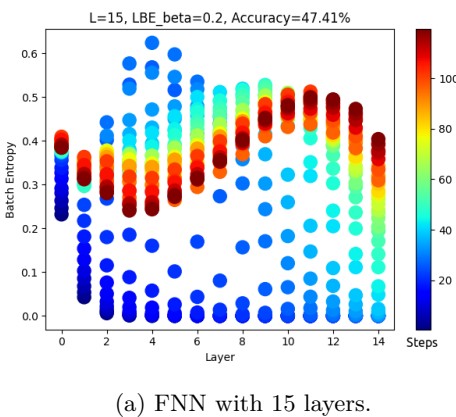
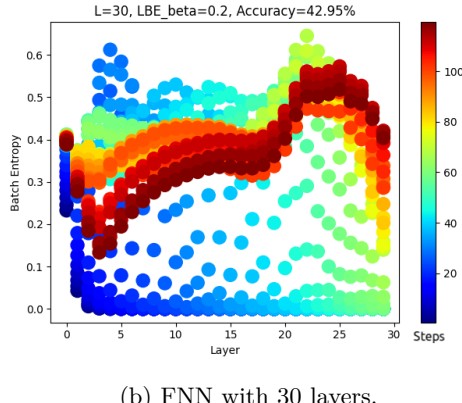

(a) FNN with 15 layers.

(b) FNN with 30 layers.

Figure 5: Flow of information through different layers of an FNN, trained on MNIST for 2 epochs with batch-entropy regularization for $\beta = 0.2$, all $\alpha^l$ are initialized with 0.5. The $x$ axis shows the layer $l$ of the network to be evaluated and the $y$ axis is its the batch-entropy. The color scheme indicates the step after which the batch-entropy was evaluated. Early training stages are blue, later training stages are shown in red.

## 4 Experimental Evaluation

We analyze our hypothesis 1 and hypothesis 2 empirically on computer vision as well as natural language processing tasks for a wide range of architectures, namely fully connected networks, residual networks, transformer models and autoencoders.

### 4.1 Setup

All experiments are implemented in PyTorch (Paszke et al., 2017) and executed on Nvidia GPUs. Wandb was used for experiment tracking (Biewald, 2020). Our source code is publicly available on GitHub[2], including sweep files for each experiment which describe precise hyperparameter ranges that we used in order to reproduce all experiments. We split the training set into training (80%) and validation (20%) and use the validation set to find a good hyperparameter setup for each method. We want to explicitly mention that the training set that we used is therefore smaller if compared against other approaches, which often use all training samples. Precise hyperparameter values for each experiment are given in the appendix. The performance results, comparing models that are trained with / without LBE, is provided on the test-set to ensure that we do not overfit the validation set through hyperparameter search. Following previous work on deep learning (Devlin et al., 2019), we report the average over 5 random restarts using different seeds. In total we used 8 different datasets for evaluation, including computer vision as well as natural language processing tasks: MNIST (LeCun & Cortes, 2010), FashionMNIST (Xiao et al., 2017), CIFAR10 and CIFAR100 (Krizhevsky et al., 2009), RTE (Bentivogli et al., 2009), MRPC (Dolan & Brockett, 2005) and CoLA (Warstadt et al., 2018). To also evaluate the proposed approach on a more challenging task, while still using only moderate computational resources, we trained a model on a subset of ImageNet (100 classes) but used only 500 images per class to increase the complexity of the task (Deng et al., 2009). We call this dataset ImageNet$_{100}$ in this paper.

### 4.2 Training Deep Vanilla Neural Networks

To further evaluate hypothesis 1, we trained "vanilla" networks with 500 layers. The accuracy, $\mathcal{L}_{CE}$ and $\mathcal{L}_{BE}$ loss during training are shown in fig. 6 for a fully connected as well as convolutional neural network. We found that such deep vanilla networks are not trainable if no $\mathcal{L}_{BE}$ was used. We also compared batch-entropy regularization against other normalization techniques as reported in appendix F

---

[2]https://github.com/peerdavid/layerwise-batch-entropy

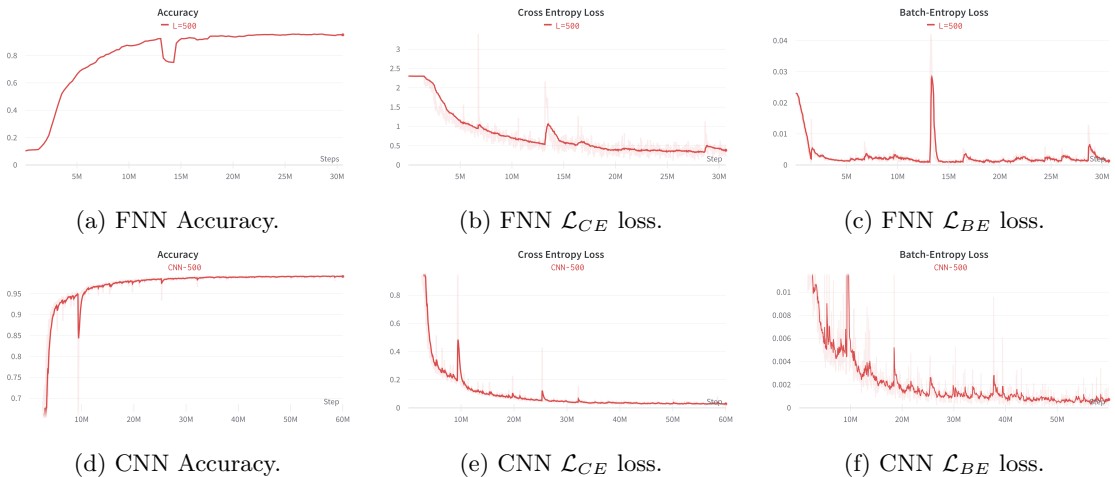

Figure 6: Shows the training curve of a "vanilla" fully connected network (FNN) and "vanilla" convolutional neural network (CNN). Both networks are 500 layers deep and both are trained on MNIST. The accuracy as well as the two loss terms ($\mathcal{L}_{CE}$, $\mathcal{L}_{BE}$) are shown.

With batch-entropy regularization, this "vanilla" fully connected network reached a test accuracy of 95.2% which shows that we can indeed train very deep vanilla fully connected networks by optimizing the information flow using $\mathcal{L}_{BE}$ regularization. The training curve is shown in fig. 6a-6c. We also tested whether batch-entropy regularization can be used to train deep vanilla convolutional neural networks. More precisely, we trained a network with 500 convolutional layers with $ReLU$ activation functions as shown in fig. 6d-6f. We did not use pooling layers, skip connections or any other normalization technique. We reached a test accuracy of 99.2% on MNIST with this setup. Specific hyperparameters can be found in the source code that we provide. This experiment shows that batch-entropy regularization can not only be used to make untrainable fully connected networks trainable, but can also be applied to other architectures such as convolutional neural networks. Another interesting finding is that for fully connected as well as convolutional networks the batch-entropy loss also increased during training, which indicates that the regularization method must be applied during the whole training process to keep the network trainable.

We are not only interested to evaluate the effect of batch-entropy regularization for very deep vanilla networks but also want to analyze the effect on other architectures with less layers. We therefore evaluate a wide range of different architectures and depths on computer vision as well as natural language processing tasks in the next experiments.

## 4.3 Fully Connected Neural Networks

In this section, we empirically evaluate the effect of layerwise batch-entropy regularization on fully connected networks of different depth. We executed a quantitative experiment and trained fully connected networks with depths from 10 to 50 and evaluated the test accuracy for each network when trained with and without $\mathcal{L}_{BE}$. All networks contain 1000 neurons per layer that are trained for 100 epochs, with a batch size of 512, using the Adam optimizer (Kingma & Ba, 2015), on MNIST (LeCun & Cortes, 2010). Weights are initialized with the method as proposed by He et al. (2015). To find a good setup for the baseline, we optimized for the learning rate $lr \in [1e-4, 5e-4, 1e-3]$ utilizing grid-search, and took the best run w.r.t the validation set. Specific hyperparameter values which were determined using grid search are provided in appendix H.1 ( table 6).

A quantitative analysis of different fully connected networks is shown in table 1. The mean accuracy of 5 runs on our test set together with the standard error is shown. First of all, it can be seen that deep networks (depth 30 - 50) can only be trained if the $\mathcal{L}_{BE}$ loss is used, where the batch-entropy of each layer is regularized in order to enable training of deep neural networks with gradient descent (hypothesis 1). For

| Depth | $\mathcal{L}_{BE}$ | Test Accuracy | Std. Error |
|---|---|---|---|
| 10.0 | No | 98.94 | 0.01 |
| | Yes | **98.95** | 0.01 |
| 20.0 | No | 97.86 | 0.60 |
| | Yes | **98.48** | 0.13 |
| 30.0 | No | 11.35 | 0.00 |
| | Yes | **92.84** | 1.86 |
| 40.0 | No | 11.35 | 0.00 |
| | Yes | **93.51** | 0.67 |
| 50.0 | No | 11.35 | 0.00 |
| | Yes | **92.93** | 1.22 |

Table 1: Comparison of training fully connected networks with or without batch-entropy regularization. All fully connected networks are trained for 100 epochs with a batch size of 512. Weights are initialized with the method as proposed by He et al. (2015). Larger numbers are written in bold.

a depth of 10, the test-accuracy is almost the same when training with and without $\mathcal{L}_{BE}$. For a depth of 20, batch-entropy regularization shows a very positive effect on the final performance of the model, even though the network is still trainable without the $\mathcal{L}_{BE}$. In this case, the variance between different runs is also heavily reduced because of the use of $\mathcal{L}_{BE}$ (0.13 compared to 0.60), which indicates that batch-entropy regularization not only helps to convert untrainable networks into trainable networks, but also has a positive effect on the training in general. Another interesting observation we can extract from appendix H.1 is that $\alpha$—found through grid search—is 0.1 for almost all depths. Only for the deepest network with 50 layers $\alpha = 0.3$. This indicates that (1) $\alpha$ values are quite consistent for a given architecture and datasets, which could help to reduce the hyperparameter search space the future runs and (2) deeper networks require a larger flow of information through each layer to reach high performance since the deepest network required a larger $\alpha$ value.

### 4.4 Autoencoders

Autoencoders are a class of unsupervised learning algorithms traditionally used for dimensionality reduction or feature learning. Autoencoders are neural networks with a bottleneck in their architecture, usually in the middle hidden layer. The network predicts the input itself, hence the bottleneck forces the network to learn a compressed representation $\mathbf{z} \in \mathbb{R}^{N_z \times 1}$ of the data, such that for input $\mathbf{x}$, the *encoder* returns $\mathbf{z} = f(\mathbf{x})$. The *decoder* reconstructs the input from such representation, $\hat{\mathbf{x}} = g(\mathbf{z})$. Assuming each $x \in \mathbf{x}$ is Bernoulli distributed, the autoencoder is trained by minimizing the binary cross-entropy between the ground-truth $\mathbf{x}$ and the reconstructed input $\hat{\mathbf{x}}$.

We trained autoencoders with depths $d \in \{1, 5, 10, 25\}$, where a depth of $d$ means that both the encoder and the decoder have $d$ hidden layers each. The dimension of the bottleneck was set to $N_z = 10$ and all hidden layers contain 256 neurons. All networks were trained for 50 epochs with a batch size of 128. The Adam (Kingma & Ba, 2015) optimizer with a waterfall schedule was used. An initial learning rate of $1e-3$ was used and every time the validation error stopped decreasing—estimated over a window of 5 epochs—the learning rate was scaled down by a factor of 0.5. Experiments were run for the MNIST (LeCun & Cortes, 2010) and FashionMNIST (Xiao et al., 2017) datasets. All images were flattened to a one-dimensional array of size 784. For evaluation we compare the similarity between the target $\mathbf{x}$ and reconstruction $\hat{\mathbf{x}}$ using the *Mean Structural Similarity Index* (MSSIM) (Wang et al., 2004).

The average MSSIM and std. error of 5 runs on the test set are given in Table 2. The first observation is that for a depth of one the performance between the autoencoders with and without batch-entropy regularization is the same, adding batch-entropy regularization has no negative (nor positive) effect on the performance of shallow autoencoders (hypothesis 2). For larger depths—$d \in \{5, 10, 25\}$—autoencoders with batch-entropy regularization outperform their counterparts without batch-entropy regularization. The performance, though, is not as high as for a shallow autoencoder. For a depth of $d = 5$ the MSSIM is still close

| Dataset | Depth | $\mathcal{L}_{BE}$ | Test MSSIM | Std. Error |
|---|---|---|---|---|
| FashionMNIST | 1 | no | **0.61** | 0.0005 |
| | | yes | **0.61** | 0.0007 |
| | 5 | no | 0.53 | 0.0138 |
| | | yes | **0.57** | 0.0009 |
| | 10 | no | 0.13 | 0.0001 |
| | | yes | **0.41** | 0.0084 |
| | 25 | no | 0.13 | 0.0002 |
| | | yes | **0.41** | 0.0059 |
| MNIST | 1 | no | **0.75** | 0.0013 |
| | | yes | **0.75** | 0.0017 |
| | 5 | no | 0.12 | 0.0005 |
| | | yes | **0.65** | 0.0071 |
| | 10 | no | 0.12 | 0.0003 |
| | | yes | **0.42** | 0.0038 |
| | 25 | no | 0.12 | 0.0004 |
| | | yes | **0.39** | 0.0168 |

Table 2: Comparison of training autoencoders with or without batch-entropy regularization. The networks are trained for 50 epochs with a batch size of 128 using the Adam optimizer. A depth of $d$ means that both the encoder and the decoder have $d$ hidden layers each. Larger numbers are written in bold.

to the best MSSIM, especially on the FashionMNIST dataset. For a depth larger than 5 we see that the network is only trainable if the $\mathcal{L}_{BE}$ regulaizer is used (hypothesis 1). Deep autoencoders with depths 5, 10, 25 and 50 and without batch-entropy regularization, all collapse at similar local optima, MSSIM = 0.12 for MNIST and MSSIM = 0.13 for FashionMNIST. The network outputs the mean value of all images in the training set $\mathcal{S}$—$1/|\mathcal{S}| \sum_{i=1}^{|\mathcal{S}|} \mathbf{x}_i$—independently of the input, which is a widely known problem of deep autoencoders (Murphy, 2012). On the other hand, autoencoders trained with the $\mathcal{L}_{BE}$ regularizer do not suffer from this problem as shown in table 2.

### 4.5   Residual Networks

In the previous experiments we have seen empirically that batch-entropy regularization can indeed convert untrainable fully connected networks, convolutional networks as well as autoencoder networks into trainable networks. In the next experiments we additionally evaluate whether batch-entropy regulariaztion has a positive or negative effect on regularizing networks with residual connections, which are known to overcome the degradation problem. More precisely, He et al. (2016) introduced residual blocks containing multiple convolutional layers that are bypassed with skip connections. The performance is further improved with batch-normalization layers and dropout just before the output layer. In this experiment, we evaluate the effect of optimizing the information flow through networks with residual connections.

We used a learning rate of 0.1 at the beginning of the training and divide it by 10 whenever it plateaus, together with a weight decay of 0.0001 and a momentum of 0.9. The network is trained on four different datasets which are MNIST (LeCun & Cortes, 2010), FashionMNIST (Xiao et al., 2017), CIFAR10 and CIFAR100 (Krizhevsky et al., 2009) using SGD with a mini-batch size of 256 (He et al., 2016). Weights are initialized with the method as proposed by He et al. (2015). To compute the $\mathcal{L}_{BE}$, we regularized the information flow of the intermediate layers within a residual block as we found empirically that regularization of the complete residual block has no effect.

The results on the test set are shown in table 3. Results indicate that generalization of residual networks benefits from batch-entropy regularization as the network trained with $\mathcal{L}_{BE}$ outperforms the baseline in 10 out of 12 cases (hypothesis 2). The standard error decreases as well for almost all evaluations indicating more stable results when executed with different seeds. Another observation is that the median of $\alpha$ increased

| Dataset | Depth | $\mathcal{L}_{BE}$ | Test Accuracy | Std. Error |
|---|---|---|---|---|
| MNIST | 26 | No | **99.68** | 0.02 |
| | | Yes | 99.67 | 0.01 |
| | 74 | No | 99.67 | 0.02 |
| | | Yes | **99.68** | 0.01 |
| | 152 | No | 99.59 | 0.02 |
| | | Yes | **99.66** | 0.03 |
| FashionMNIST | 26 | No | **94.33** | 0.06 |
| | | Yes | 94.25 | 0.06 |
| | 74 | No | 93.89 | 0.09 |
| | | Yes | **94.18** | 0.09 |
| | 152 | No | 93.33 | 0.13 |
| | | Yes | **93.55** | 0.17 |
| CIFAR10 | 26 | No | 89.40 | 0.03 |
| | | Yes | **89.41** | 0.08 |
| | 74 | No | 89.70 | 0.20 |
| | | Yes | **89.89** | 0.22 |
| | 152 | No | 88.16 | 0.31 |
| | | Yes | **88.54** | 0.19 |
| CIFAR100 | 26 | No | 62.85 | 0.10 |
| | | Yes | **63.12** | 0.05 |
| | 74 | No | 63.95 | 0.30 |
| | | Yes | **64.63** | 0.15 |
| | 152 | No | 62.47 | 1.40 |
| | | Yes | **64.68** | 0.49 |
| ImageNet$_{100}$ | 26 | No | 51.08 | 0.22 |
| | | Yes | **51.47** | 0.25 |
| | 74 | No | 51.84 | 0.21 |
| | | Yes | **52.28** | 0.20 |
| | 152 | No | 50.33 | 1.92 |
| | | Yes | **52.45** | 0.28 |

Table 3: Comparison of training different residual networks with or without batch-entropy regularization. All residual networks are trained for 100 epochs with a batch size of 256. Weights are initialized with the method as proposed by He et al. (2015). Larger numbers are written in bold.

as the complexity of the dataset increased (appendix H.2): For MNIST, FashionMNIST, CIFAR10, and CIFAR100 the median of $\alpha$ for all different depths is 0.5, 1.0, 1.0, and 1.5 respectively. It seems reasonable that more complex data (e.g. more classes, more color channels, etc.) would require more information throughput and therefore, a higher batch-entropy.

## 4.6 Transformer Models

Transformer models are a class of models which implement self-attention layers followed by fully connected layers in order to find correlations between different parts of input tokens (e.g. words in a sentence) (Devlin et al., 2019). Those networks are pre-trained in a self-supervised fashion on a huge text corpus such as wikipedia. After the pre-training they are fine-tuned on a given downstream task. We will now evaluate whether batch-entropy regularization is also beneficial for this type of models, which are already pre-trained. Different from previous experiments on computer vision tasks, transformers are mostly used for natural language processing. More precisely, we trained a BERT$_{Base}$ with 12 layers and BERT$_{Large}$ model with 24

| Dataset | Model | $\mathcal{L}_{BE}$ | Test Perf. | Std. Error |
|---------|-------|------|-----------|-----------|
| RTE | $\text{BERT}_{Base}$ | no | **67.42** | 0.89 |
|  |  | yes | 67.34 | 1.54 |
|  | $\text{BERT}_{Large}$ | no | 69.76 | 1.20 |
|  |  | yes | **70.78** | 0.87 |
| MRPC | $\text{BERT}_{Base}$ | no | 88.58 | 0.70 |
|  |  | yes | **89.13** | 0.88 |
|  | $\text{BERT}_{Large}$ | no | 89.06 | 0.37 |
|  |  | yes | **89.28** | 0.55 |
| CoLA | $\text{BERT}_{Base}$ | no | 55.63 | 0.97 |
|  |  | yes | **56.70** | 1.04 |
|  | $\text{BERT}_{Large}$ | no | 59.39 | 0.71 |
|  |  | yes | **60.13** | 0.63 |

Table 4: Comparison of training with or without batch-entropy regularization for $\text{BERT}_{Base}$ with 12 layers and $\text{BERT}_{Large}$ with 24 layers. All networks are pre-trained on a large-scale text corpus (Devlin et al., 2019) and fine-tuned with a batch size of 32. Larger numbers are written in bold.

layers on the RTE (Bentivogli et al., 2009), MRPC (Dolan & Brockett, 2005) and CoLA (Warstadt et al., 2018) dataset.

Following Devlin et al. (2019), we fine-tuned networks with a batch-size of 32 and a learning rate that we found through a grid-search $\in [1e-5, 3e-5, 5e-5]$. Similar to the findings about residual networks, batch-entropy regularization improves the performance of trained models for transformer models on natural language processing tasks (hypothesis 2). Only for $\text{BERT}_{Base}$ trained on RTE, the model was slightly worse when including $\mathcal{L}_{BE}$. For the rest of cases, there was an improvement when including $\mathcal{L}_{BE}$ regularization. Of special interest is the MPRC, where the $\text{BERT}_{Base}$ with $\mathcal{L}_{BE}$ had a higher score (89.13), than the $\text{BERT}_{Large}$ without $\mathcal{L}_{BE}$ (89.06). These latter results show that those pre-trained models are not fully exploited yet and the downstream-performance can be further improved without the need of additional pre-training.

We finalise our analysis by exposing some interesting facts regarding the differences between residual networks and transformer models. For residual networks, we found that the performance improved most when the $\alpha$ value—which regularizes the information flow—is large ($\approx 1.0$). On the other hand, for transformer models, a small $\alpha$ value ($\approx 0.3$) improved their performance the most, which would indicate that the compression and reduction of information are beneficial. Our insights on this observation are as follows: Transformer models implement a pooling layer just before the classifier which pools all output tokens (e.g. 128, 256, or 512) except for the first token. Compressing as much information as possible into the first token therefore would help for classification. A complete and precise evaluation of this hypothesis is out of scope for this paper and is left for future work.

## 5 Conclusion and Discussion

In this paper, we analyzed the flow of information through deep neural networks. We introduced the batch-entropy to approximate the amount of information that is propagated through each individual layer. A positive batch-entropy is necessary at each layer of the network in order to optimize a given objective function with gradient descent correctly as shown in section 3. Through dimensionality reduction and the analysis of the loss surface, we found that whenever the information flow through a network is zero, the gradient is also zero almost everywhere.

The batch-entropy regularization term presented in this work optimizes the flow of information through the network by regularizing the batch-entropy of each layer individually. We showed through extensive experimental evaluation that networks not trainable at first, could be trained thanks to the inclusion of the proposed regularizer. Blot et al. (2018) also proposed a regularizer in order to improve the intermediate representations of single samples $x$ but they calculate the entropy of a sample conditioned to the correspond-

ing label $y$. The authors showed that a model that is invariant to many transformations will produce the same representation for different inputs, which improves the performance of the model. Instead of penalizing layer-transformations, we add a regularization term to ensure a positive flow of information through each layer. Our empirical evaluation also showed that this regularization term has a positive effect on the performance of a trained model for a wide range of tasks and architectures. Specifically, such positive effect was consistent across different networks, such as fully connected, residual, autoencoder, and transformer networks applied to computer vision and natural language processing tasks.

The results of our experimental evaluation are very promising and we think it can serve to improve future deep learning research, even so, we would like to finish our analysis with some open questions. First, even though the $\alpha$ value—which regularizes the information flow—can be consistently set for a network architecture trained on a given task, the value is currently determined through a grid search. One solution worth of further analysis is to obtain a good initial value for $\alpha$ from an analysis of the entropy of training data, this is a line of work on which we are currently working. Second, the batch-entropy is currently computed through different samples of a mini-batch and can therefore only be derived whenever the batch size is larger than one. If a batch size of one is used, similar methods as gradient accumulation can be better suited, which is worth investigating in future work. Finally, in the case of transformer models, we found that compression of information improves scores, which is a strong indicator that the current pooling layer is not optimal w.r.t a given downstream task. Further research could include not just regularizing networks with $\mathcal{L}_{BE}$, but also a novel pooling layer to improve scores. To end with a final positive outcome, we found a positive effect of batch-entropy regularization on generalization and in our future work we would like to additionally investigate, whether this same method has also a positive effect on the adversarial robustness of a trained model.

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

# A Distribution of Individual Neurons

The distribution of output values of 25 randomly selected neurons for 128 different output values is shown in fig. 7. The neural network was trained on the MNIST dataset.

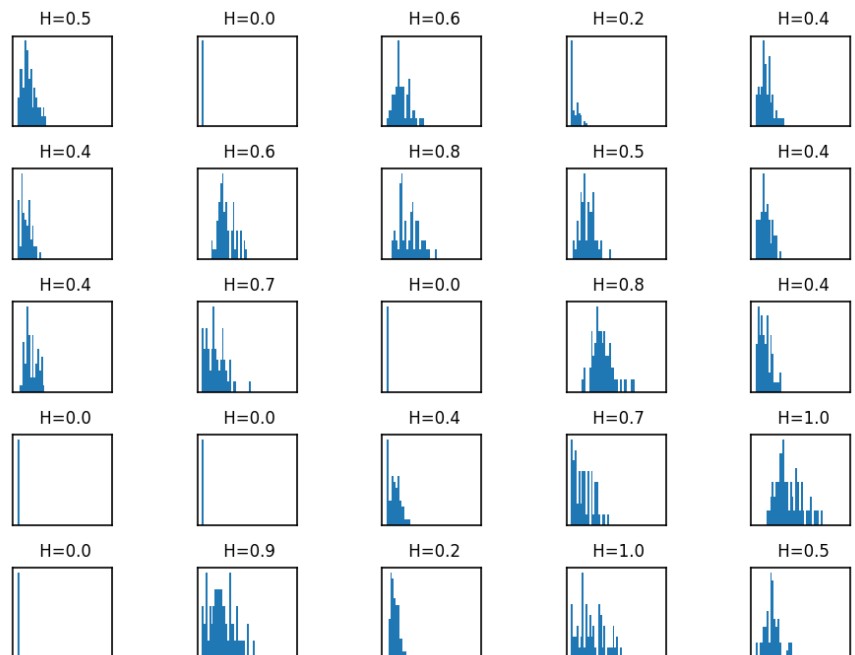

Figure 7: Histogram of 1024 output values of several neurons from a fully connected network trained on MNIST. The x-axis shows the output-value for each neuron and the y-axis depicts the number of times that output-value was observed. On the top of each plot, the estimated batch-entropy eq. (3) is shown.

# B Information Theoretic Perspective on the Entropy of Subsequent Layers

We denote a random variable with a capital letter, e.g., $X$. A single event is denoted with a lower case letter $\mathbf{x} \in X$ (note that $\mathbf{x}$ can be multi-dimensional). The probability density function of a continuous random variable is given be $p(\mathbf{x})$ and the mutual information between random variables $X$ and $Y$ is given by $I(X;Y)$.

A neural network can be interpreted as a *Markov Chain* (Tishby & Zaslavsky, 2015; Shwartz-Ziv & Tishby, 2017), where each hidden layer is a (stochastic) map of the previous layer. The training data is represented by random variables $X$ and $Y$, for the objects and labels respectively, and have a joint probability distribution $p(x, y)$ with $I(X; Y) > 0$. The output of a hidden layer $l$ is represented by random variable $T_l$ and the output of the network is represented by random variable $\hat{Y}$. A neural network with $L$ hidden layers forms the following *Markov Chain*:

$$Y \longleftrightarrow X \longrightarrow T_1 \longrightarrow T_2 \longrightarrow \cdots \longrightarrow T_L \longrightarrow \hat{Y}, \tag{10}$$

where $Y \leftrightarrow X$ indicates that the factorization of $p(x, y)$ is not known. The *Data Processing Inequality* (DPI) (Cover & Thomas, 2006) gives us the following inequalities:

$$I(Y; X) \geq I(Y; T_1) \geq I(Y; T_2) \geq \ldots \geq I(Y; T_L) \geq I(Y; \hat{Y}). \tag{11}$$

Which implies that our estimate $\hat{Y}$ can at most contain as much information about $Y$ as $X$. It also implies that each hidden layer $T_l$ can at most contain as much information about $Y$ as the previous hidden layer $T_{l-1}$.

In the worst case all, information about the input $X$ gets lost and we will have $I(X; T_l) = 0$ for some hidden layer $l$. Given that the only information $T_l$ can contain about $Y$ is through $X$, we also have $I(Y; T_l) = 0$. The DPI tells us that $I(X; T_l) \geq I(X; T_{l+1}) \geq \ldots \geq I(X; \hat{Y})$, hence $I(X; T_l) = I(X; T_{l+1}) = \ldots = I(X; \hat{Y}) = 0$. The output $\hat{Y}$ contains no information about the input $X$ (and $Y$).

A standard neural network is deterministic, which is as a special case of the problem described above. The mutual information between input $X$ and layer $l$ can be written as:

$$I(X; T_l) = H(T_l) - H(T_l \mid X), \tag{12}$$

where $H(T_l)$ is the (differential) entropy of layer $l$ and $H(T_l \mid X)$ is the conditional (differential) entropy of layer $l$ given input $X$, i.e., the entropy that cannot be explained by input $X$. The conditional entropy represents external information, e.g., noise, that is being added during the processing steps from input $X$ to layer $l$. For deterministic neural networks it is safe to assume no external information is added, i.e., $H(T_l \mid X) = 0$. Given $I(X; T_l) = 0$, this implies $H(T_l) = H(T_l \mid X) = 0$. In other words, if no information besides the input is added to the network and a layer $l$ contains no information about the input, it must be zero entropy.

## C  Derivation of the Gradient if the Batch-Entropy is Zero

Without loss of generality we can assume that labels are one-hot encoded vectors $\mathbf{y}$ that are uniformly distributed w.r.t different classes in our mini-batch $\mathcal{B}$. If we assume that the batch-entropy is zero, it follows that

$$\frac{\partial \mathcal{L}(\mathcal{B})}{\partial \mathbf{W}^L} = \frac{1}{|\mathcal{B}|} \sum_{b=1}^{|\mathcal{B}|} \frac{\partial \mathcal{L}(\mathbf{x}_b)}{\partial \mathbf{W}^L} \qquad \text{Gradient of mini-batch}$$

$$= \frac{1}{|\mathcal{B}|} \sum_{b=1}^{|\mathcal{B}|} (g(\mathbf{x}_b) - \mathbf{y}_b) \; a^L(\mathbf{x}_b)^T \qquad \text{Gradient as defined in section 3.1}$$

$$= \frac{1}{|\mathcal{B}|} \sum_{b=1}^{|\mathcal{B}|} (\mathbf{g} - \mathbf{y}_b) \; a^L(\mathbf{x}_b)^T \qquad H^L = 0 \implies g(\mathbf{x}_0) = g(\mathbf{x}_1) = ...g(\mathbf{x}_{|\mathcal{B}|}) = \mathbf{g}$$

$$= \frac{1}{|\mathcal{B}|} \sum_{b=1}^{|\mathcal{B}|} (\mathbf{g} - \mathbf{y}_b) \; \mathbf{a}^T \qquad H^L = 0 \implies a^L(\mathbf{x}_0) = a^L(\mathbf{x}_1) = ...a^L(\mathbf{x}_{|\mathcal{B}|}) = \mathbf{a}$$

$$= \left( \mathbf{g} - \frac{1}{|\mathcal{B}|} \sum_{b=1}^{|\mathcal{B}|} \mathbf{y}_b \right) \mathbf{a}^T \qquad \text{Rewrite}$$

$$= \left( \mathbf{g} - \frac{1}{c} \mathbf{1} \right) \mathbf{a}^T, \qquad \text{Uniformly distributed batches w.r.t different classes of } \mathbf{y}_b$$

where $\mathbf{1}$ is a vector with components 1 of dimension $c$ for a dataset consisting of $c$ classes.

We next evaluate the gradient if not only the cross-entropy is used to train a model, but also batch-entropy regularization as proposed in eq. (7):

$$\frac{\partial \mathcal{L}}{\partial \mathbf{W}^l} = 0 + \mathcal{L}_{CE} \frac{\partial \mathcal{L}_{BE}}{\partial \mathbf{W}^l} + 0 \qquad \text{Assumption that } \frac{\partial \mathcal{L}_{CE}}{\partial \mathbf{W}^l} = 0$$

$$= \mathcal{L}_{CE} \frac{\partial}{\partial \mathbf{W}^l} \left( \frac{\beta}{L} \sum_{k=0}^{L} \mathcal{L}_{BE}^k \right)$$

$$= \mathcal{L}_{CE} \frac{\partial}{\partial \mathbf{W}^l} \left( \frac{\beta}{L} \mathcal{L}_{BE}^l \right)$$

$$= \mathcal{L}_{CE} \frac{\beta}{L} \frac{\partial \mathcal{L}_{BE}^l}{\partial \mathbf{W}^l}.$$

## D   Interpretation of Batch-Entropy Regularization.

The following illustration shows why networks with batch-entropy regularization are trainable whenever the batch-entropy is zero.

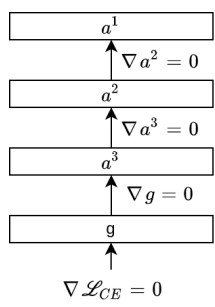
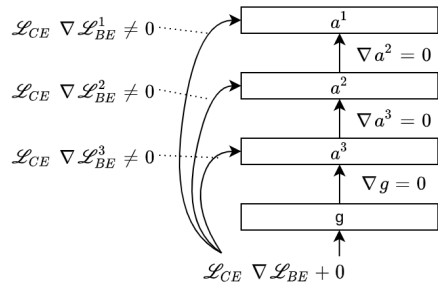

(a) Backpropagation without batch-entropy regularization

(b) Backpropagation with batch-entropy regularization

Figure 8: Comparison of backpropagation without (a) and with (b) $\mathcal{L}_{BE}$ regluarization whenever $H^L = 0$.

## E Loss Surfaces of $\mathcal{L}_{CE}$, $\mathcal{L}_{BE}$ and Accuracy

In this appendix we show the loss surfaces for the $\mathcal{L}_{BE}$ as well as the $\mathcal{L}_{CE}$ loss after different stages of training. It can be seen that weights of the network are never pushed into regions that can be optimized w.r.t. the objective function, if the batch-entropy regularization is not used as shown in fig. 9, fig. 10, and fig. 11.

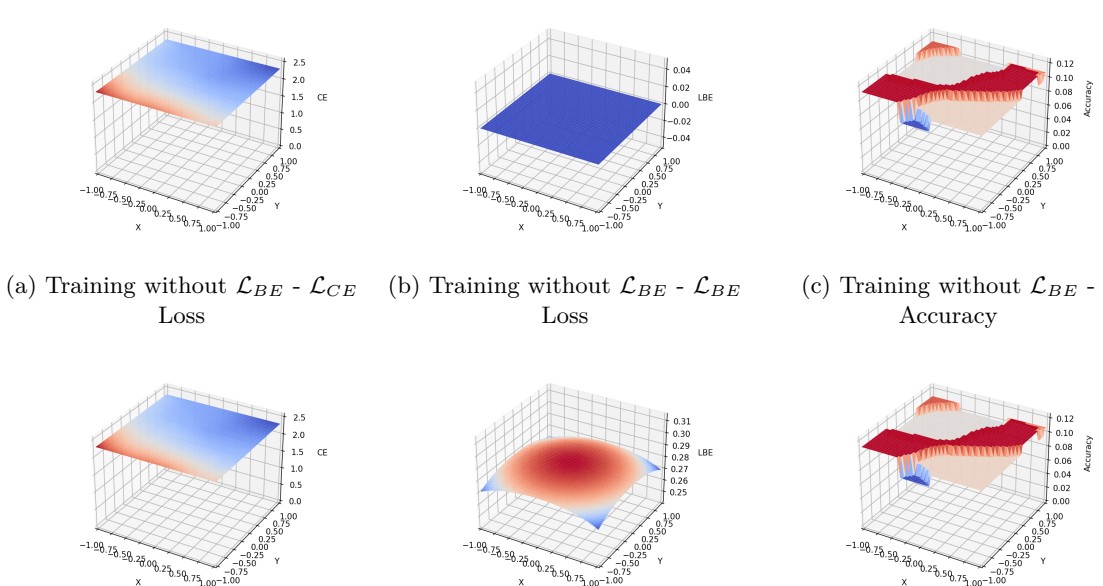

(a) Training without $\mathcal{L}_{BE}$ - $\mathcal{L}_{CE}$ Loss

(b) Training without $\mathcal{L}_{BE}$ - $\mathcal{L}_{BE}$ Loss

(c) Training without $\mathcal{L}_{BE}$ - Accuracy

(d) Training with $\mathcal{L}_{BE}$ - $\mathcal{L}_{CE}$ Loss

(e) Training with $\mathcal{L}_{BE}$ - $\mathcal{L}_{BE}$ Loss

(f) Training with $\mathcal{L}_{BE}$ - Accuracy

Figure 9: Different loss surfaces and accuracy for a network with 30 layers at the beginning of training on MNIST. The $\mathcal{L}_{BE}$ Loss points away from the local minimum leading to a network that can be optimized with cross entropy.

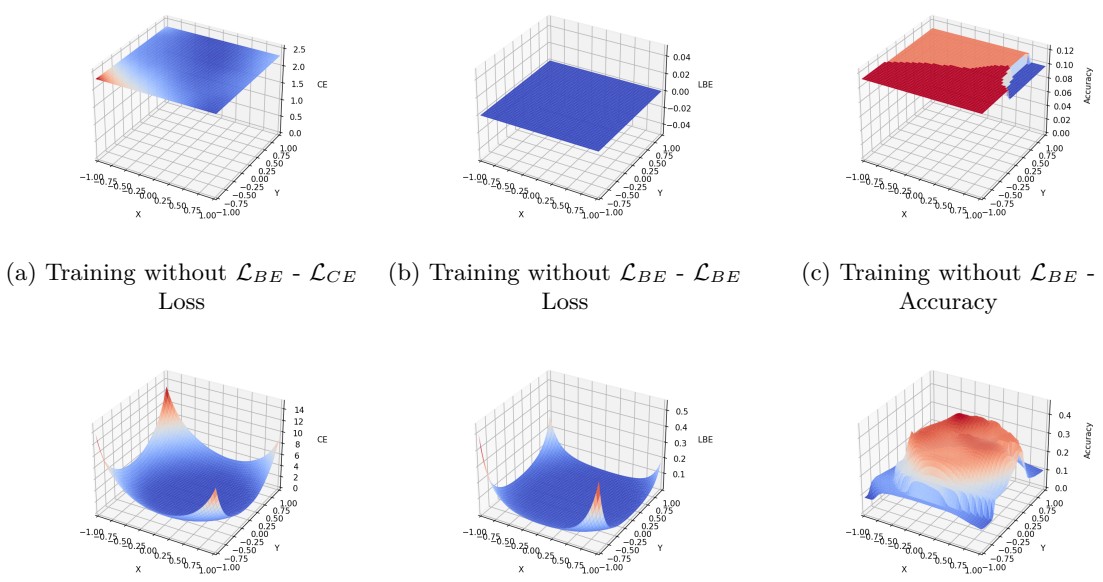

(a) Training without $\mathcal{L}_{BE}$ - $\mathcal{L}_{CE}$ Loss

(b) Training without $\mathcal{L}_{BE}$ - $\mathcal{L}_{BE}$ Loss

(c) Training without $\mathcal{L}_{BE}$ - Accuracy

(d) Training with $\mathcal{L}_{BE}$ - $\mathcal{L}_{CE}$ Loss  (e) Training with $\mathcal{L}_{BE}$ - $\mathcal{L}_{BE}$ Loss  (f) Training with $\mathcal{L}_{BE}$ - Accuracy

Figure 10: Different loss surfaces and accuracy for a network with 30 layers trained on MNIST. After training for 1k steps the $\mathcal{L}_{BE}$ is close to the cross entropy loss such that the focus of optimization is on minimizing the real objective function (cross entropy) rather than the information flow ($\mathcal{L}_{BE}$).

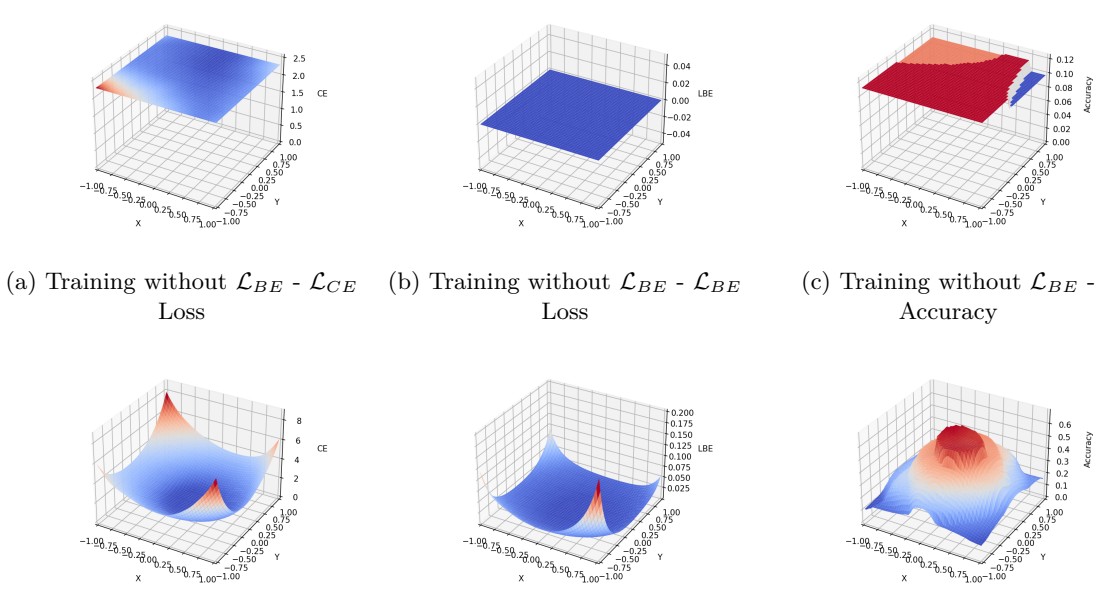

(a) Training without $\mathcal{L}_{BE}$ - $\mathcal{L}_{CE}$ Loss

(b) Training without $\mathcal{L}_{BE}$ - $\mathcal{L}_{BE}$ Loss

(c) Training without $\mathcal{L}_{BE}$ - Accuracy

(d) Training with $\mathcal{L}_{BE}$ - $\mathcal{L}_{CE}$ Loss  (e) Training with $\mathcal{L}_{BE}$ - $\mathcal{L}_{BE}$ Loss  (f) Training with $\mathcal{L}_{BE}$ - Accuracy

Figure 11: Different loss surfaces and accuracy for a network with 30 layerstrained on MNIST. After training for 2k steps the $\mathcal{L}_{BE}$ is close to the cross entropy loss such that the focus of optimization is on minimizing the real objective function (cross entropy) rather than the information flow ($\mathcal{L}_{BE}$).

## F  Comparison with Normalization Methods

Normalization methods are used to improve the trainability of deep neural networks. In this section we will show that four commonly used normalization methods—batch normalization (Ioffe & Szegedy, 2015), layer normalization (Ba et al., 2016), weight normalization (Salimans & Kingma, 2016) and self-normalizing neural networks (Klambauer et al., 2017b)—cannot convert untrainable networks into trainable networks.

For each normalization method we train four "vanilla" fully connected network of different depths on the MNIST dataset. Each network has 500 layers with 1000 neurons per layer and the ReLU activation function is used—except for the self-normalizing neural network which uses the SELU activation function. Each network is trained for 500 epochs using the cross entropy loss and the Adam optimizer. We evaluated different learning rates and only show the results for the best performing learning rate—highest accuracy on the test set. The results are shown in Table 5. We were not able to train a "vanilla" fully connected neural network of 500 layers with any of the normalization methods. As opposed to our LBE regularizer, which does allow us to train such a deep network.

| Normalization Method | Test Accuracy |
|---|---|
| Batch Normalization | 11.74 |
| Layer Normalization | 11.35 |
| Self-Normalizing Neural Networks | 11.35 |
| Weight Normalization | 11.35 |
| LBE Regularization | 95.20 |

Table 5: Comparison of different normalization methods with the LBE regularizer. Results are obtained using a fully connected neural network of 500 layers and 1000 neurons per layer. Different learning rates and seeds were tested and only the best performing model—in terms of test accuracy—is shown.

## G  Hyperparameter $\alpha$

The selection of the hyperparameter $\alpha$ is critical for batch-entropy regularization in order to successfully train the network. In the experimental section 4 we executed a grid search in order to find optimal values for $\alpha$ (values are reported in appendix H). In this section correlations between different hyperparameter and $\alpha$ for fully connected networks as well as residual networks are shown.

**Fully Connected Networks**  We trained fully connected networks with learning rates $\in [3e-5, 5e-5, 1e-4]$, depths $\in [10, 20, 30, 40, 50, 500]$ and evaluated which value for $\alpha$ (with $\alpha \in [0.5, 1.0, 1.5]$) produced the result with highest performance. We report the mean of five runs with different seeds. Results for experiments with different depths as well as learning rates are shown in fig. 12.

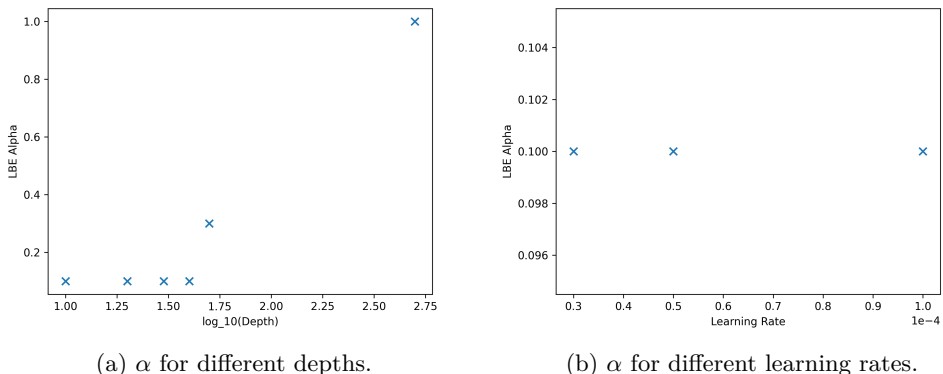

(a) $\alpha$ for different depths.

(b) $\alpha$ for different learning rates.

Figure 12: Shows the optimal selection of $\alpha$ for fully connected networks trained on MNIST.

In fig. 12a it can be seen that larger $\alpha$ values are preferable as the depth of the network increases. It is reasonable that the flow of information through all the layers must be larger for deeper networks.

On the other hand it can also be seen that for all different learning rates a $\alpha = 0.1$ works best which indicates that $\alpha$ does not depend on this hyperparameters and can be selected independently.

We trained residual networks with depths $\in [26, 74, 152]$ on MNIST, FashionMNIST, CIFAR10, and CIFAR100 and evaluated which value for $\alpha$ (with $\alpha \in [0.5, 1.0, 1.5]$) produced the result with the highest performance for five different runs. The results are shown in fig. 12.

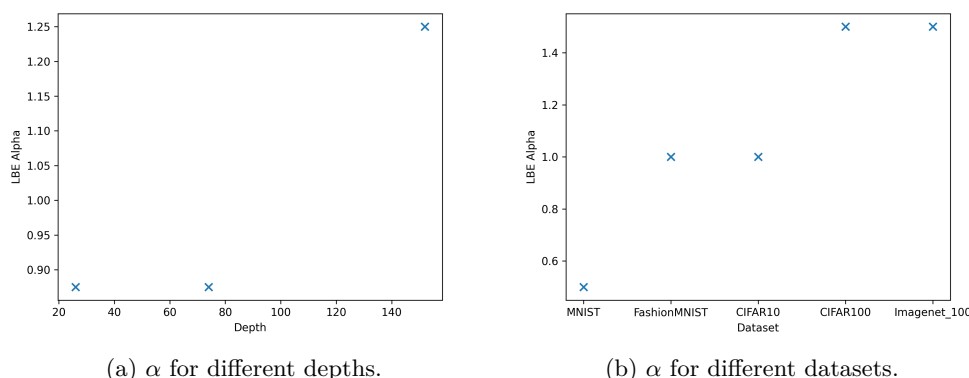

(a) $\alpha$ for different depths.

(b) $\alpha$ for different datasets.

Figure 13: Shows the optimal selection of $\alpha$ for residual networks.

Similar to our findings in fig. 12a for fully connected networks, we can see the same pattern also for residual networks in fig. 13a as the optimal alpha value increases with the depth of the network.

Additionally we report in fig. 12b the optimal value for different datasets. It can be seen that as the complexity of the dataset increases, a larger $\alpha$ value is desirable. For example for MNIST, which are 10 different classes containing grayscale images the optimal $\alpha$ value is smaller when compared to the more complex colored images of CIFAR10. It can be seen that as the number of classes increases e.g. from CIFAR10 to CIFAR100 a larger $\alpha$ helps further to improve the performance of the network.

# H Hyperparameter Tuning LBE

In this appendix, we report the hyperparameters $\alpha$ and $\beta$ that are used for batch-entropy regularization. We found the values through a grid search. Additionally, we report the validation and test accuracy for each setup.

## H.1 Fully Connected Networks

| Depth | Alpha | Beta | Val. Accuracy | Test Accuracy | Std. Error |
|-------|-------|--------|---------------|---------------|------------|
| 10.0 | 0.0 | 0.0000 | 97.96 | 98.94 | 0.01 |
| | 0.1 | 0.0005 | 97.97 | **98.95** | 0.01 |
| 20.0 | 0.0 | 0.0000 | 95.83 | 97.86 | 0.60 |
| | 0.1 | 0.0001 | 96.83 | **98.48** | 0.13 |
| 30.0 | 0.0 | 0.0000 | 11.17 | 11.35 | 0.00 |
| | 0.1 | 0.0010 | 88.70 | **92.84** | 1.86 |
| 40.0 | 0.0 | 0.0000 | 11.17 | 11.35 | 0.00 |
| | 0.1 | 0.0005 | 88.10 | **93.51** | 0.67 |
| 50.0 | 0.0 | 0.0000 | 11.17 | 11.35 | 0.00 |
| | 0.3 | 0.0001 | 87.45 | **92.93** | 1.22 |

Table 6: Evaluation of fully connected networks.

## H.2 Residual Networks

.

| Dataset | Depth | Alpha | Beta | Val. Accuracy | Test Accuracy | Std. Error |
|---------|-------|-------|------|---------------|---------------|------------|
| MNIST | 26 | 0.0 | 0.000 | 99.51 | **99.68** | 0.02 |
| | | 0.5 | 0.010 | 99.51 | 99.67 | 0.01 |
| | 74 | 0.0 | 0.000 | 99.47 | 99.67 | 0.02 |
| | | 0.5 | 0.005 | 99.48 | **99.68** | 0.01 |
| | 152 | 0.0 | 0.000 | 99.39 | 99.59 | 0.02 |
| | | 1.0 | 0.001 | 99.42 | **99.66** | 0.03 |
| FashionMNIST | 26 | 0.0 | 0.000 | 93.57 | **94.33** | 0.06 |
| | | 0.5 | 0.005 | 93.69 | 94.25 | 0.06 |
| | 74 | 0.0 | 0.000 | 93.32 | 93.89 | 0.09 |
| | | 1.0 | 0.010 | 93.39 | **94.18** | 0.09 |
| | 152 | 0.0 | 0.000 | 92.76 | 93.33 | 0.13 |
| | | 1.5 | 0.001 | 92.99 | **93.55** | 0.17 |
| CIFAR10 | 26 | 0.0 | 0.000 | 89.49 | 89.40 | 0.03 |
| | | 1.0 | 0.005 | 89.59 | **89.41** | 0.08 |
| | 74 | 0.0 | 0.000 | 89.71 | 89.70 | 0.20 |
| | | 0.5 | 0.005 | 90.04 | **89.89** | 0.22 |
| | 152 | 0.0 | 0.000 | 88.20 | 88.16 | 0.31 |
| | | 1.0 | 0.005 | 88.66 | **88.54** | 0.19 |
| CIFAR100 | 26 | 0.0 | 0.000 | 62.24 | 62.85 | 0.10 |
| | | 1.5 | 0.005 | 62.53 | **63.12** | 0.05 |
| | 74 | 0.0 | 0.000 | 63.02 | 63.95 | 0.30 |
| | | 1.5 | 0.001 | 64.03 | **64.63** | 0.15 |
| | 152 | 0.0 | 0.000 | 61.32 | 62.47 | 1.40 |
| | | 1.5 | 0.005 | 63.82 | **64.68** | 0.49 |
| Imagenet$_{100}$ | 26 | 0.0 | 0.000 | 48.65 | 51.08 | 0.22 |
| | | 1.5 | 0.010 | 48.78 | **51.47** | 0.25 |
| | 74 | 0.0 | 0.000 | 49.03 | 51.84 | 0.21 |
| | | 1.0 | 0.001 | 49.42 | **52.28** | 0.20 |
| | 152 | 0.0 | 0.000 | 47.97 | 50.33 | 1.92 |
| | | 1.5 | 0.010 | 49.47 | **52.45** | 0.28 |

Table 7: Evaluation of residual networks.

### H.3 Autoencoder

| Dataset | Model | Depth | Alpha | Beta | Val. MSSIM | Test MSSIM | Std. Error |
|---|---|---|---|---|---|---|---|
| FashionMNIST | AE | 1 | 0.0 | 0.0 | 0.61 | **0.61** | 0.0005 |
| | | | 1.5 | 0.1 | 0.61 | **0.61** | 0.0007 |
| | | 5 | 0.0 | 0.0 | 0.54 | 0.53 | 0.0138 |
| | | | 1.5 | 0.2 | 0.58 | **0.57** | 0.0009 |
| | | 10 | 0.0 | 0.0 | 0.13 | 0.13 | 0.0001 |
| | | | 1.5 | 0.2 | 0.41 | **0.41** | 0.0084 |
| | | 25 | 0.0 | 0.0 | 0.13 | 0.13 | 0.0002 |
| | | | 2.0 | 0.5 | 0.41 | **0.41** | 0.0059 |
| MNIST | AE | 1 | 0.0 | 0.0 | 0.75 | **0.75** | 0.0013 |
| | | | 1.5 | 0.1 | 0.75 | **0.75** | 0.0017 |
| | | 5 | 0.0 | 0.0 | 0.11 | 0.12 | 0.0005 |
| | | | 2.5 | 0.5 | 0.64 | **0.65** | 0.0071 |
| | | 10 | 0.0 | 0.0 | 0.11 | 0.12 | 0.0003 |
| | | | 1.5 | 0.1 | 0.41 | **0.42** | 0.0038 |
| | | 25 | 0.0 | 0.0 | 0.11 | 0.12 | 0.0004 |
| | | | 1.5 | 0.1 | 0.40 | **0.39** | 0.0168 |

Table 8: Evaluation of autoencoders.

### H.4 Transformer

| Dataset | Model | Alpha | Beta | Val. Performance | Test Performance | Std. Error |
|---|---|---|---|---|---|---|
| RTE | $BERT_{Base}$ | 0.0 | 0.000 | 68.42 | **67.42** | 0.89 |
| | | 0.5 | 0.010 | 69.00 | 67.34 | 1.54 |
| | $BERT_{Large}$ | 0.0 | 0.000 | 70.38 | 69.76 | 1.20 |
| | | 0.2 | 0.010 | 72.17 | **70.78** | 0.87 |
| MRPC | $BERT_{Base}$ | 0.0 | 0.000 | 86.62 | 88.58 | 0.70 |
| | | 0.2 | 0.005 | 86.80 | **89.13** | 0.88 |
| | $BERT_{Large}$ | 0.0 | 0.000 | 87.04 | 89.06 | 0.37 |
| | | 0.3 | 0.005 | 88.34 | **89.28** | 0.55 |
| CoLA | $BERT_{Base}$ | 0.0 | 0.000 | 58.48 | 55.63 | 0.97 |
| | | 0.2 | 0.050 | 58.81 | **56.70** | 1.04 |
| | $BERT_{Large}$ | 0.0 | 0.000 | 63.22 | 59.39 | 0.71 |
| | | 0.3 | 0.005 | 64.31 | **60.13** | 0.63 |

Table 9: Evaluation of transformers.

