# OpenReview forum: "Improving the Trainability of Deep Neural Networks through Layerwise Batch-Entropy Regularization"
_TMLR — Accepted by TMLR_

### Review · Reviewer_H1f1 · 2022-05-20

**Summary Of Contributions:**

This paper proposes a new strategy to train deep neural networks without shortcuts. It adds a batch-entropy regularization to optimize the information flow. With the regularization strategy, vanilla fully connected networks can be well trained. The authors conduct experiments on multiple datasets to validate the effectiveness.

**Broader Impact Concerns:**

No ethical implications.

**Requested Changes:**

-The authors are required to validate the proposed method on larger datasets such as ImageNet.

**Strengths And Weaknesses:**

Strengths

- Training deep neural networks is a critical problem in deep learning. A vanilla network without shortcuts occupies less memory. It is also easy to implement for fast inference.

-The proposed batch-entropy regularization is reasonable to make the gradients stable. Without any architectural tweak, an over 500 layers neural network can be well trained.

-The proposed method can work well on multiple architectures, such as fully connected network transformers.

Weakness:

-The authors only conduct experiments on small datasets, such as CIFAR10, CIFAR100, FashionMNIST, and MNIST. The classification tasks on these small datasets are usually easy to achieve high performance. Besides, it also has a large difference from practical scenarios.

- A network with 152 layers can be well trained. It is interesting to see whether a convolutional neural network over 1000 layers can be trained.

---

> ### Author Response · Authors · 2022-06-08
> **Answer from Authors**
>
> First of all, we want to thank the reviewer for the effort put into detailed comments that helped us much to improve the quality of our paper and the confirmation that training deep vanilla networks with batch-entropy regularization could be interesting for future research too. We adapted the paper accordingly and answer each question raised in the review:
>
> > The authors only conduct experiments on small datasets, such as CIFAR10, CIFAR100, FashionMNIST, and MNIST. The classification tasks on these small datasets are usually easy to achieve high performance. Besides, it also has a large difference from practical scenarios.
>
> We agree with the reviewer and updated the paper with the results different trained networks with different depths on a subset of 100 classes from Imagenet, which allows us to use a more complex dataset while being able to run more than 30 different necessary runs in due time. As this decreases the complexity of the task, we artificially increased the difficulty by reducing the number of training samples to only 500 samples per class. Precise values are reported in the revision of the paper, please not we got an improvement on all depths with batch-entropy regularization. The biggest improvement (+2.45%) was for ResNet-152.
>
>
> > A network with 152 layers can be well trained. It is interesting to see whether a convolutional neural network over 1000 layers can be trained.
>
> We followed this interesting suggestion and trained a deep vanilla CNN. We did not use skip connections, batch normalization or any other architectural tweak. We can already confirm that this vanilla CNN is trainable if batch-entropy regularization is used and report the training curve in the revision of the paper. More precisely, with this architecture we reached a test accuracy of 99.2%.

---

### Review · Reviewer_U3Z5 · 2022-05-22

**Summary Of Contributions:**

The paper proposes a measure for the information flow through each layer of the network. The proposed measure is the layerwise batch entropy, which is zero when the variance of the output of a layer is zero and large when the variance is large. The main observation of the paper is that when this entropy is small, deeper networks are not-trainable. Then the paper proposes a regulariser to control this entropy and show that deeper networks are trainable when this entropy is controlled.

**Requested Changes:**

* Critical - The main paper should be shortened to a much smaller length in my opinion.
* Critical - The theoretical section, right now, does not appear to have a major contribution for this paper, in my opinion. So, I think one way to improve it would be to give insights on how to choose $\alpha$, though I realise that is a difficult problem.
* Critical - The experimental results need to be strengthened. I think the authors need to make a clear case for when to use this regulariser is useful. Without it, I will be tending towards rejection.

**Strengths And Weaknesses:**

Strength
* I liked Figure 2, which shows that different layers indeed has different entropies. Further, the change in entropy from one layer to the next seems to be continuous, which is rather interesting though I am not sure if this has any significance.
* The style of identifying a property that makes networks untrainable, and then fixing it to increase trainability is very interesting to me and I applaud the authors for that.

Weakness
* As per TMLR guidelines, there is no page limit but the length of the paper should determine how rigorous the review should be. In short, the length should justify the contribution. In this paper, I am strongly of the opinion that it does not. The main text is about 15 pages long, which is rather long for the contributions.
* One of the main points seem to be that Hypothesis 1, which states that a network is not trainable with gradient descent of there is a layer where the entropy is zero, which essentially means that when the output of the layer is invariant to the input. But that is quite obvious that if the output does not depend on the input, the gradient will be zero as it just a constant function. Then the network will indeed be untrainable. Could the authors please clarify if this is an incorrect interpretations ?
* Regarding the experimental contribution, first, the choice of $\alpha$ seems important but is not discussed properly, especially how do you choose $\alpha$ for different datasets and networks ? How does the performance change with different choices of the hyper-parameter i.e. how brittle is it ?
* Finally, the experimental results are not convincing at all. In CIFAR10 and MNIST, there is no separation between using the regulariser or not using the regulariser. In fact, shallower networks actually perform better. in CIFAR100, the difference is less than 1% and similar with FasionMNIST. For residual layers, the observation is same as for  fully connected layers.Even for BERT, there is no real difference (within 1%). For auto-encoders there appears to be some difference for deeper networks, which is perhaps the most positive experimental contribution.

---

> ### Author Response · Authors · 2022-06-08
> **Answer from Authors**
>
> We want to thank the reviewer for the effort put into detailed comments. It was our main goal to provide interesting insights and a new approach to train deep vanilla networks.
>
> > One of the main points seem to be that Hypothesis 1, which states that a network is not trainable with gradient descent [..] Could the authors please clarify if this is an incorrect interpretations ?
>
> This interpretation is not completely correct because the gradient is not zero at the beginning of the training. We found (theoretically and empirically) that the gradient points into a direction that is independent of its label "y" and therefore, training pushes weights into a local minimum where each neuron outputs values of "1/c" as argued in the paper (described in section 3.2).
>
> > Regarding the experimental contribution, first, the choice of \alpha seems important but is not discussed properly, especially how do you choose for different datasets and networks?
>
> Many different hyperparameters of regularization methods are often found through HPO methods such as random-search or grid searches. We followed this approach and searched \alpha through a grid search as we describe in the experimental section. Precise values are added to the appendix and supplementary material. Following the suggestion, we added a section to the revised paper on \alpha which shows different choices of \alpha. We also describe some findings on \alpha in section 4.2, 4.3 as well as 4.5.
>
> > How does the performance change with different choices of the \alpha hyper-parameter i.e. how brittle is it ?
>
> In general, we found that selecting \alpha in order to transform an untrainable network into a trainable network is not brittle - alpha should simply be increased proportionally to depth. If the value for \alpha is too small, the network is not trainable.
>
> > Finally, the experimental results are not convincing at all. [..]
>
> First of all, we want to clarify that our focus is not on achieving state-of-the-art results on some benchmarks as this is not the focus of our paper, nor of TMLR if we understood the guidelines correctly. The main goal of our work is to introduce a regularization term such that untrainable networks are transformed into trainable networks during the training process, without the need for special activation functions or other architectural tweaks. In the revised version of the manuscript, we have tried to make the focus of our work clear. We have also added new experiments where we show that we can also train very deep convolutional networks which further motivates our claim (test accuracy of 99.2% on MNIST). A comparison against other normalization techniques and Imagenet_100 are also added to the revised paper.
>
>
> > Critical - The main paper should be shortened to a much smaller length in my opinion.
>
> Our feelings are a bit mixed as other reviewers claimed that "Presentation and writing of the paper is clear, well-structure and concise" and also the action editor asked for additional content. We reduced the length of the theoretical section but as the action editor and other reviewers asked for additional content, the overall length did not decrease. We are open to consider suggestions on parts of the paper that could be considered unnecessary.
>
> > Critical - The theoretical section, right now, does not appear to have a major contribution for this paper, in my opinion. So, I think one way to improve it would be to give insights on how to choose \alpha, though I realise that is a difficult problem.
>
> We have reduced the theoretical section as requested, but we respectfully disagree in its importance as we introduce not just batch-entropy but also introduce batch-entropy regularization in this section, which is very relevant to our work and the rest of the paper. Regarding on how to choose \alpha, we are doing research in the direction to derive a precise value for \alpha (Future work).  We also added a new section to the appendix where we show how to choose \alpha and we describe some interesting findings in section 4.2, 4.3 and 4.5.
>
> > Critical - The experimental results need to be strengthened. I think the authors need to make a clear case for when to use this regulariser is useful. Without it, I will be tending towards rejection.
>
> We appreciate knowing that in the present writing, our regularizer may not be clear enough for readers, and to clarify this important fact we adapted as follows:
> - We changed the title as we believe that the reviewer found an extremely important weakness. This should make it clear that batch-entropy is mainly to improve the trainability of networks.
> - We also adapted the abstract, the introduction, and the related work section where we now also added more references to motivate our fairly simple approach further.
> - Finally, we also distinguished our work from other normalization techniques as well as signal theory-inspired work in the introduction and through additional experiments.

---

### Review · Reviewer_LvEn · 2022-06-01

**Summary Of Contributions:**

The authors present an approach to regularize information
flow through neural network layers. This leads to a differentiable
loss term that is added throughout training.


**Broader Impact Concerns:**

Ok.

**Requested Changes:**

Requested changes are formulated "the authors should [..]" in my statements above.

**Strengths And Weaknesses:**


General comments: The approach is novel however strongly related
to normalization as the proposed term is just log variance.
The method is not well-embedded into prior literature, especially
it misses out on the signal propagation theory. It is unclear
whether this approach is relevant because the experiments are
mostly done on small scale datasets and far from SOTA.

Positive aspects/strengths:
- Presentation and writing of the paper is clear, well-structure and concise
- Experiments try to tackle a wider variety of tasks to show the generality of the method
(however, mostly small datasets and far from current SOTA)


Major comments/weaknesses:
a) Novelty: The approach is novel but closely related to all normalization techniques
and signal propagation theory. The main quantitity, Eq (2) and Eq(3),
is just the log variance in the layer. The theory revolves around
H^l being zero, which indicates that the variance of the neuron activations
is zero. It is well-established in the community that this leads
to untrainable networks. This is the reason why normalization techniques
have been developed (references [1]-[4]).
There is also a huge body of work on signal propagation in random networks
(references [5]-[8]) which also focus, for example, on keeping the variance
of the neuron (pre-) activations in a certain range.
The authors should embed their work into other related works and clearly
state their novel contributions, for example at the end of the introduction section.

b) Relevance: The experiments are performed on small scale datasets such
as MNIST, CIFAR10, CIFAR100 and in a performance range of methods from
many years ago. Because of the substantial closely related methods, such
as all normalization techniques, that were used to train networks on
much larger datasets (ImageNet, etc) and at higher performance ranges, it
is unclear whether this method is relevant at all. There are many possible
ways in which variance (or information) can be maintained or transformed
through neural network layers, such that a clear improvement over those
related techniques should be demonstrated. Improvement over un-normalized
deep neural networks are hardly relevant since it is established that specialized
techniques are necessary to train them.
The authors should demonstrate the effectiveness of their approach/method
on larger datasets and with architectures operating at usual performance ranges
close to or at state-of-the-art. Methods compared should include the different
normalization techniques ([1]-[4]) and methods from signal propagation theory (e.g. [5]-[8].


Minor:
Eq (3): some error as i does not appear in the formula on the right



References:
[1] Ioffe, S., & Szegedy, C. (2015, June). Batch normalization: Accelerating deep network training by reducing internal covariate shift. In International conference on machine learning (pp. 448-456). PMLR.
[2] Ba, J. L., Kiros, J. R., & Hinton, G. E. (2016). Layer normalization. arXiv preprint arXiv:1607.06450.
[3] Salimans, T., & Kingma, D. P. (2016). Weight normalization: A simple reparameterization to accelerate training of deep neural networks. Advances in neural information processing systems, 29.
[4] Klambauer, G., Unterthiner, T., Mayr, A., & Hochreiter, S. (2017). Self-normalizing neural networks. Advances in neural information processing systems, 30.
[5] Xiao, L., Bahri, Y., Sohl-Dickstein, J., Schoenholz, S., & Pennington, J. (2018, July). Dynamical isometry and a mean field theory of cnns: How to train 10,000-layer vanilla convolutional neural networks. In International Conference on Machine Learning (pp. 5393-5402). PMLR.
[6] Lee, Jaehoon, Yasaman Bahri, Roman Novak, Samuel S. Schoenholz, Jeffrey Pennington, and Jascha Sohl-Dickstein. "Deep neural networks as gaussian processes." arXiv preprint arXiv:1711.00165 (2017).
[7] Martens, J., Ballard, A., Desjardins, G., Swirszcz, G., Dalibard, V., Sohl-Dickstein, J., & Schoenholz, S. S. (2021). Rapid training of deep neural networks without skip connections or normalization layers using Deep Kernel Shaping. arXiv preprint arXiv:2110.01765.
[8] Blumenfeld, Y., Gilboa, D., & Soudry, D. (2020, November). Beyond signal propagation: is feature diversity necessary in deep neural network initialization?. In International Conference on Machine Learning (pp. 960-969). PMLR.

---

> ### Author Response · Authors · 2022-06-10
> **Answer from Authors**
>
> First of all, we want to thank the reviewer for the effort put into detailed comments and we are thankful for the list of related papers that help us to further distinguish and motivate our approach.
>
> Before addressing and clarifying the concerns of the reviewer we would like to motivate and reason for submitting to TMLR. It is our goal to publish interesting findings and insights rather than pushing state-of-the-art results on some benchmarks, which seems to be the aim of TMLR. It is our view that the major weaknesses (novelty and relevance) given by the reviewer contradict the goals and guidelines of TMLR and should therefore not be considered major weakness. The reviewer guidelines state:
>
>     "Crucially, it should not be used as a reason to reject work that isn't considered “significant” or “impactful” because it isn't achieving a new state-of-the-art on some benchmark. Nor should it form the basis for rejecting work on a method considered not “novel enough”, as novelty of the studied method is not a necessary criteria for acceptance."
>
> Nevertheless, the reviewer does provide important and valuable feedback that will improve our paper and we would like to express our gratitude for that once more.
>
> > Novelty: The approach is novel but closely related to all normalization techniques and signal propagation theory. [...] The authors should embed their work into other related works and clearly state their novel contributions
>
> We are thankful for this hint and we changed the introduction as well as related work section in order to clearly separate our work from the normalization techniques mentioned [1]-[4] and we also compare it against the signal propagation theory papers. Additionally, we provide new experimental results in the appendix to further support our claim and to distinguish our work.
>
> We also executed another experiment with vanilla convolutional networks and compared our approach against [5] as proposed by the reviewer. We compared a deep vanilla CNN trained with batch-entropy regularization (99.2% on MNIST) and tried to compare it against [5]. Unfortunately, up-to-date code versions of [5] (https://github.com/yl-1993/ConvDeltaOrthogonal-Init and https://github.com/JiJingYu/delta_orthogonal_init_pytorch) only work with up to 150 layer CNNs. We also found after measuring the batch-entropy that the flow of information drops after 150 layers which made the network untrainable.
>
>
> > The theory revolves around H^l being zero, which indicates that the variance of the neuron activations is zero. It is well-established in the community that this leads to untrainable networks. This is the reason why normalization techniques have been developed (references [1]-[4]).
>
> We politely ask the reviewer for the references that would support such a claim. To the best of our knowledge, the normalization techniques [1]-[4] are mainly developed to accelerate and improve training of deep networks, which is a very different line of research to ours, where we transform untrainable networks into trainable networks to be able to train very deep vanilla networks.
>
> In order to clarify this aspect, we executed additional experiments where we tested whether the mentioned normalization techniques can transform untrainable into trainable networks. More precisely we compared [1]-[4] against batch-entropy regularization for training a vanilla fully connected network with 500 layers as described in our paper and we found that none of the normalization techniques [1]-[4] can be used to train such a deep network successfully. We tested many different hyperparameters but found not a single setup that leads to a model with performance higher than chance.
>
>
> > Relevance: The experiments are performed on small-scale datasets such as MNIST, CIFAR10, CIFAR100 [...] Methods compared should include the different normalization techniques ([1]-[4]) and methods from signal propagation theory (e.g. [5]-[8].
>
> In the revised version of the paper we have tried to clarify the main purpose of batch-entropy regularization is to improve trainability of neural networks and not another benchmark-performance-breaking regularizer, which we believe falls in the objetives of TMLR. Even so, we understand that we should have done a better job conveying our contributions and to this aim, we have updated the title, abstract as well as introduction in order to make our contribution clear.
>
> Following the reviewer suggestion, we also added additional experiments to show that we can not only train deep fully connected networks, but also deep convolutional networks,  we have added additional experiments using a more complex dataset based on Imagenet, added a comparison against [1]-[4] and distinguished our work from [5]-[8] in the introduction as well as related work section.

---

### Decision · Action_Editors · 2022-07-28

**Recommendation:** Accept as is

**Comment:**

Overall this is a borderline submission. I do appreciate however the author's rebuttal and the improvements to the paper which I believe address most of the reviewer's concerns.

The reviewer's original concerns can be summarised in two key points:
1. Related work: Relation to existing work needs to be expanded and clarified substantially. Low theoretical novelty.
2. Relevance: Experiments are only on small scale datasets, where high performance does not require very deep models. So, the proposed method is not tested on regimes where it would have impact.

The change of the focus towards "trainability of untrainable deep models" does sound more appropriate for the contribution. However, in the end of the day, the reason we want to train deeper models is still because we want to apply them to larger scale problems/datasets. So this change of focus does sound a bit like a way to avoid more criticisms of type 2) above. I hope this method will be tested in more interesting problems/datasets in future work.

While I initially agreed with points 1) and 2) and also that the draft was too long for the contribution; The revised manuscript satisfactory addresses point (1,2) in my view in light of TMRL guidelines. While the theoretical contribution is somewhat straightforward and partially addressed in previous work on layer-normalisation (eg. see refs [1-4, 5, 8] pointed out by Rev. LvEn), the newly added experiments  (Appendix F, Table 5) show that the idea does bring something new to the picture relative to existing normalisation methods. It is a common pattern in DL literature that some mathematically trivial ideas can have a big practical impact simply because they enable training of larger models or new model classes.